# Formalising Human-in-the-Loop: Computational Reductions, Failure Modes, and Legal–Moral Responsibility

**Maurice Chiodo**
University of Cambridge
mcc56@cam.ac.uk

**Dennis Müller**
University of Cologne
dennis.mueller@uni-koeln.de

**Paul Siewert**
University of Cambridge
pks50@cam.ac.uk

**Jean-Luc Wetherall**
DeepFin Research
jlw95@cantab.ac.uk

**Zoya Yasmine**
University of Oxford
zoya.yasmine@law.ox.ac.uk

**John Burden**
University of Cambridge

## Abstract

We use the notion of oracle machines and reductions from computability theory to formalise different Human-in-the-loop (HITL) setups for AI systems, distinguishing between trivial human monitoring (i.e., total functions), single endpoint human action (i.e., many-one reductions), and highly involved human–AI interaction (i.e., Turing reductions). We then proceed to show that the legal status and safety of different setups vary greatly. We present a taxonomy to categorise HITL failure modes, highlighting the practical limitations of HITL setups. We then identify omissions in UK and EU legal frameworks, which focus on HITL setups that may not always achieve the desired ethical, legal, and sociotechnical outcomes. We suggest areas where the law should recognise the effectiveness of different HITL setups and assign responsibility in these contexts, avoiding human 'scapegoating'. Our work shows an unavoidable trade-off between attribution of legal responsibility, and technical explainability. Overall, we show how HITL setups involve many technical design decisions, and can be prone to failures out of the humans' control. Our formalisation and taxonomy opens up a new analytic perspective on the challenges in creating HITL setups, helping inform AI developers and lawmakers on designing HITL setups to better achieve their desired outcomes.

## 1 Introduction

Human-in-the-loop (HITL)—the practice of embedding human oversight into a computational machine, such as an Automated Decision Making System (ADMS) or AI system—is frequently invoked as a safeguard to ensure AI safety and accountability (Green, 2022). We show that the effectiveness of HITL for satisfying both safety and regulatory requirements hinges critically on the specifics of the ADMS's design and implementation. We present a novel formalisation of HITL setups as computational reductions (§2.1). We analyse those corresponding to total functions, many-one reductions, and Turing reductions, showing that each leads to vastly different safety outcomes. Existing legal frameworks, such as Article 22 of GDPR (2016) and Article 14 of EU AI Act (2024) only consider HITL under a simplistic view, and do not account for the variety of setups we show possible. And so, we clarify the various outcomes (§2), failure modes (§3), and legal and moral responsibilities (§4) associated with each of our setups. We further demonstrate an inherent, ever present tension when using HITL, whereby setups with increased human interpretability and explainability hinder clear attribution of responsibility, and vice versa (§4.3). Ultimately, HITL is no panacea to the problem of ADMS safety, and we identify six concrete suggestions for its effective use (§5).

Although introducing HITL is increasingly a strategic choice to improve sociotechnical systems (Grønsund & Aanestad, 2020), our focus is not on optimising the average-case performance, but rather on ensuring safety, reliability, and robustness. We are primarily concerned with preventing ADMS-related harms by recognising that different degrees of human involvement in HITL

setups carry varied technical and ethical implications that should be recognised by the law. Our perspective therefore asks when a HITL setup represents genuine and meaningful human control of an ADMS, and encompasses its moral, legal and political ideals of keeping 'social control' over technology (Abbink et al., 2024). Our manuscript responds to the question of when, if ever, should we use HITL? And how can we select HITL setups in order to minimise *harm* and *risk* and thus ensure *safety* (all defined in A.1)? (cf. Pillar 1 in Chiodo & Müller (2025).)

Much terminology exists for human–ADMS interactions. HITL setups have humans actively integrated into the ADMS's deployment cycle (we do not cover human input in training or development). Human-on-the-loop (HOTL) has humans acting as supervisors, intervening only when necessary. Setups where the ADMS operates without direct human intervention are termed Human-out-of-the-loop (HOOTL). Human-in-Command (HIC) has humans determine the high-level functioning of the ADMS. Meaningful Human Control (MHC) was introduced to study what influence a human should have on the execution of an action and what levels of cognitive and moral awareness they need. Further discourse on these terms is given in A.2. Our work demonstrates how different HITL setups can be generalised and formalised using computability theory, unifying these related but disparate concepts. Such formalisation creates a general framework to analyse HITL setups across varied contexts, and provides a consistent way for regulators to examine HITL involvement where required by law, thereby also reducing the ability for companies to introduce tokenistic HITL setups.

Though intended to protect society, HITL setups can end up protecting the ADMS instead, with the human becoming the 'moral crumple zone' taking on accountability and enabling potentially faulty ADMSs to persist (Elish, 2019). Hence, a closer scrutiny of HITL setups and their failure modes is necessary. The lack of a well-defined formal meaning of what HITL can and does involve, and why HITL may be beneficial in sociotechnical systems, is a recognised problem. And yet, having a HITL setup is frequently presented as a critical safety measure, even in high-risk domains (with supporting details in A.3). Recognising the different degrees to which humans are, and should be, involved in HITL setups, we present a classification of these using computational reductions. Our primary contributions are threefold. In §2 we give a novel formalisation of HITL setups which unifies various known HITL categorisations and identify a fundamental distinction within HITL setups. In §3 we give a taxonomy of failure modes and ethical concerns for HITL setups and find they are correlated with our distinction. And in §4 we consider what HITL setups the law requires and how responsibility is assigned when they fail. In doing so, we contextualise the failure modes and categories of HITL setups to highlight ways laws can be improved to ensure safety and efficacy. We also uncover an unavoidable trade-off between responsibility and explainability in HITL setups.

## 2 COMPUTATIONAL REDUCTIONS FOR HITL SETUPS

This section aims to characterise computationally distinct HITL setups, using the concept of reductions. An **oracle machine** is a deterministic automaton $T^\bullet$ on a fixed language $W$ with a work tape, an extra 'oracle tape', and some states marked as 'oracle states'. For any function $f : W \to W$, $T^\bullet$ gives rise to a 'machine with oracle' $T^f$, whose computations proceed as usual, but which whenever it enters an oracle state has the content $w \in W$ of the oracle tape (instantly) replaced by $f(w)$; whenever $T^\bullet$ halts, it outputs the oracle tape. This definition is reminiscent of the notions of $\exists$-non-determinism and random automata, in that $T^\bullet$ by itself defines, for any input, a computation tree that contains all possible behaviours depending on the behaviour in the oracle states (respectively, the non-deterministic states or the random states). The difference is in semantics, as the result of one computation is not defined by some general property of the computation tree, but by 'plugging in' $f$ and thus resolving all the choices with $f$. A discussion of oracle machines can be found in Soare (1987, Chapter III, Section 1); we adopt here a variation of the terminology from van Melkebeek (2000, Section 2.4). Our analysis is concerned with how the computational power of $T^f$ varies with $T^\bullet$, rather than with $f$. The most salient distinction for us is that between machines which only call $f$ once, and those which call $f$ an unbounded number of times. These correspond to two different notions of 'reduction' between functions $f$ and $g$: if there is an oracle machine $T^\bullet$ such that $T^f$ computes $g$, one says $g$ is **Turing-reducible** to $f$. If moreover $T^\bullet$ is such that it halts immediately after its first call of the oracle, then one says $g$ is **many-one reducible** to $f$.

Observe that an oracle call may be rather trivial. $T^\bullet$ might ignore the content of the oracle tape, in which case the choice of $f$ does not matter. Even if $T^\bullet$ reads/writes from the oracle tape, its

computational result might not depend on these steps. However, our concern lies with the *function* defined by $T^f$, not the machine itself (even though one could examine it to increase computational transparency). Therefore, we are interested more so in what we define as a **real query**, which we say occurs at an oracle call if a fork exists at that point in the computation tree *and* not all branches have the same set of possible outputs. The former of these ensures the oracle can still have some meaningful impact on the computation, and the latter ensures we avoid an 'all roads lead to Rome' scenario where the machine can take multiple computational paths all leading to the same output.

## 2.1 HITL SETUPS AS A COMPUTATIONAL REDUCTION

We propose describing human–machine systems in terms of this formalism. Any human–machine computational process uses algorithmic components at some points and human interventions at others. We hence conceive of a HITL setup as a type of Human-Based Computation (Yuen et al., 2009), in that the machine assigns particular tasks or problems to one or more humans to solve. Our perspective is different to how the term is typically used. We are not interested in large-scale human knowledge, but rather in the specialised knowledge of individuals. And the human is not the subordinate labourer of the machine or a simple assistant, but rather is in symbiosis with the entire system. We now examine this to see what computational synergy can exist between human and machine.

The architecture of a decision-making process including the algorithmic specifications corresponds to the oracle machine $T^\bullet$. The function $f$ is provided by the human: at certain points in the process, human *judgment* on values and data enters the computation, corresponding to $T^\bullet$ calling the oracle, which we denote as a **human query**. We do not assume the human is 'correct' (see §3.1 for HITL failure modes), or even deterministic (for supporting details on how we see the human's role, see B.1). Here we include human interventions that are otherwise uninvited by the machine; computationally, it makes no difference what precipitated the human input (though in reality this still needs to be considered for evaluating safety). In response to a query, we also allow the human to write an 'emergency stop' symbol to the oracle tape, denoted by **!**; whenever the machine reads ! on the tape it halts immediately with no output. Note that the possibility to halt after a query with no output does not count towards the set of computational outcomes when determining if a query is real.

If an oracle machine is set up to never ask a human query, and humans have no way to intervene or stop its operation, this corresponds to a HOOTL from §1; we will not cover this further. If an oracle machine is set up to ask human queries, but none are real queries, then we denote this as a **human trivially monitoring the loop**, abbreviated to **trivial monitoring**. This corresponds to a HOTL setup from §1, but where the human is only able to stop the process; note they may do so at any stage in the computation, even if not explicitly asked. Computationally, in a trivial monitoring setup, the human is not affecting the computational steps of the machine in any meaningful way, and serves only to prevent the machine from continuing its computation. It is 'trivial' only in the computational sense, while still playing a crucial safety role, as argued by Crootof et al. (2023, p. 448) when discussing human sign-off requirements by 'positioning human(s) as end-of-the-loop gatekeepers'. This means that $T^\bullet$ defines a total function independent of the human. The human can only decide whether to terminate the machine before it completes its computation (an 'ignorance is bliss' scenario). If an oracle machine is instead set up to ask precisely one real human query and then immediately halt, giving the answer to that real query as its output, then we denote this as a **human at the end of the loop**, abbreviated to **endpoint action**. Here, the machine does some computation itself, then hands over to the human to perform the rest. If the oracle machine is set up to always ask potentially unbounded (and at least two) real queries (i.e., a number not bounded as a total computable function of the input), we denote this as a **human involved in the loop**, abbreviated to **involved interaction**. Practically, this is most resembling the scenario of human and machine working together, where the machine and human engage in a game of computational ping-pong: the machine does some work, hands over to the human who does some more and then hands back to the machine, and so forth. For example, a 'creative' collaboration with a Large Language Model (LLM) where the human prompts the LLM to produce an initial response (or ask a question) which the human uses to determine the next detail they give the LLM, and so on. In endpoint actions and involved interactions, we also allow the human to stop the machine at any time. Supporting details and diagrams of these setups are in B.2. Note this is a *theoretical* framing, and in any physical setup a human will naturally have limitations such as their reaction and response time, maximum amounts of data they can comprehend, computational steps they might not be able to see or 'reach', limits on how many queries they can answer, etc. Such limitations are covered later in §3.

Thus, in an *involved interaction*, the oracle machine Turing-reduces the computation to the human, but *does not* many-one reduce it. In an *endpoint action*, the oracle machine many-one reduces the computation to the human, but it *does not* define a total function. And in *trivial monitoring*, the human can terminate the machine, but not otherwise influence its computation. Underscoring the *agency* of the human (its ability to actively impact the system), our setup types are determined by the *shortest* potential human-machine interaction path in the computational tree, not the longest; supporting details are given in B.3. We explain with supporting details in B.4 why we choose not to explicitly study the myriad of intermediate reduction types, as endpoint actions and involved interactions give the 'simplest' and 'most complicated' ways a human can interact with a machine. However, we give further discourse in D.6 on how our remaining analysis in this manuscript applies to such intermediate reduction types, as well as ways to formalise them into HITL setups.

As an example, consider a route-planning machine in a HITL setup with the human driver of a car. It may demonstrate trivial monitoring (presenting one route for the human to accept or reject), or endpoint action (presenting the human several routes to choose their preferred one), or involved interaction (where the human is fully involved in the process, from early choices of where/when to travel, up to the latter optimisation of different route types, number of stops, etc.); B.5 gives further discourse on this. With growing reliance on human queries, the agency of the human and therefore the access of the machine to human values increases, resulting in decisions becoming more adjusted to human needs.

## 2.2 WHY ARE WE CONSIDERING THESE REDUCTIONS?

Computationally, a Turing reduction between functions is viewed as weaker than a many-one reduction, because we fix the computational problem first, and then ask what oracles that problem reduces to. Our considerations with HITL setups (supporting details in B.6) are the opposite; we take a *fixed oracle* (a human), and see which problems can be solved with that oracle (B.1 gives supporting details on the sense in which the human is 'fixed'). Thus, in terms of constructing optimal HITL setups, an involved interaction allows us to solve the most problems with a given human. We thus propose that the HITL setups with the greatest potential to achieve human agency, alignment, safety, transparency, and thus overall reliability, are those with the greatest reliance on human input, i.e., involved interactions (with supporting details in B.7). The more real queries made, the more we have a human *in* the loop (rather than a HOTL). In existing work, Andersen & Maalej (2024) describe 10 *design patterns* for HITL setups, with three centred around HITL during deployment/inference. The first is the 'Recommendation System' where a human makes the final decision based on the machine output. The second is 'Active Moderation' where, for a set of tasks, humans perform them in the cases where the machine cannot do so reliably. The third option 'Thumbs up or Thumbs down' sees the human either accept or correct the machine output. These are all endpoint actions or trivial monitorings. Notably, none of Andersen & Maalej (2024)'s patterns describe interactions where the task is aborted rather than corrected—effectively requiring the human to be as capable as the machine at the initial task (rather than being able to 'merely' recognise an error). In §4 we connect this to Meaningful Human Control.

In a trivial monitoring setup, the human is not involved in the computation between the machine starting and finishing its work. And so in an opaque 'black box' computational process, such as many ADMSs, one may have no idea how it was carried out. However, one can begin to 'unmask' the black-box if the machine asks real queries, as each precipitates a human-interpretable question giving *some* information about what the machine is currently 'doing' at that point in the computation. The more real queries, the more effective this unmasking is. In an endpoint action setup, the machine reveals *one* computational step at the very end, giving *some* insight into its workings. And in an involved interaction setup, there may be many real queries, revealing insight at many points throughout the computation. Of course, there may still be 'black-box' computation between these real queries. But rather than one *big* black box, the process appears as many *smaller* black boxes connected at the points where a human provides input. The machine–human 'ping-pong' can be viewed as a *chain of computations* (with supporting details and illustrations in B.8). Thus, in an involved interaction setup we argue that these smaller black boxes increase *explainability* of what went into determining the final output. This chain is relevant again in §4.3 where we discuss responsibility within HITL setups.

## 2.3 DETERMINING HITL SETUP TYPES IN PRACTICE

So how do we *identify* a HITL setup as trivial monitoring, endpoint action, or involved interaction, and why use such formalism? B.9 gives supporting details on why this is important, yet difficult. *Technically*, showing non-existence of a 'simpler' reduction is hard, as it cannot be done by example but instead requires deep analysis of the computational process. *Legally*, showing the human will do something meaningful is hard, as asking the human a series of 'pointless' questions may violate the legal principles of 'meaningful'. And *morally*, showing the HITL setup is not simply a facade masking a less-involved process is also hard, as one needs to ensure the machine does not simply ignore answers the human gives (or worse: inverts answers). Overall, our classification of HITL setups into trivial monitoring, endpoint action, and involved interaction outlines what is *possible* with HITL. But identifying these setups, from the technical, legal, and moral perspectives above, is another challenge altogether. We propose that some of the 'burden of proof' be shifted onto developers, such as obliging them to demonstrate and document various aspects of their ADMS; we argue (with supporting details in B.9) that this makes identification of the setup type a much more manageable problem.

Our route-planning example in §2.1 raises questions about the *practicalities* of HITL setups. In the endpoint action scenario, the machine could have instead asked the human a fixed finite series of questions such as 'Prioritise speed or efficiency?', 'Maximum acceptable distance between fuel stations?', etc., and done computation between each, to find a route. This would *appear* to be an involved interaction (actually a bounded truth table reduction; see supporting details in B.10). However, these could be encapsulated by instead asking the human to chose from a (long) list presented at the end; an endpoint action. But in *practical* terms, asking a human 20 binary questions is much more feasible than presenting $2^{20}$ options. Computational aspects are *one* consideration in HITL setups; the abilities and limitations of humans are another—they do not operate like abstract oracles. But, to understand how human input feeds into a HITL setup, and 'design out' the failure modes presented in the next section, we must have first understood how the human is involved from a computational perspective.

## 3 HITL FAILURE MODES

### 3.1 CHARACTERISTICS OF EFFECTIVE HITL SETUPS, AND TAXONOMY OF FAILURE MODES

According to Sterz et al. (2024), necessary and sufficient conditions for effective (general) oversight of AI are given when the human overseer has 1) an adequate understanding of the system, 2) the self-control to act on their judgment, 3) the power to intervene effectively, and 4) intentions aligned with the oversight goals. They, and others, note that successful oversight depends on many user-specific attributes, including technical skills, domain knowledge, general attitudes towards technology, especially those related to interpreting AI outputs, as well as attributes of the human–machine setup (cf. Sterz et al. (2024); Sudeeptha et al. (2024); Langer et al. (2025b)). The arguments presented in the literature remain at a comparatively abstract level, and do not distinguish by HITL setup. Overall, as our reductions in §2.1 show, the functioning of HITL setups does not rely on the design of the machine alone, nor on the characteristics of the human overseer, but on how they are put into relation with each other. This complex sociotechnical relationship allows us to turn the theoretical and empirical insights from the literature into practice, and analyse potential failure modes. We present our own taxonomy centred around five main failure categories, giving a partial breakdown of each:

**1. Failure of the machine components.** These include: Unexpected inputs or outputs, problematic machine evolution or self-adaptation, biased or erroneous outputs, and other unexpected behaviour.
**2. Failure of the process and workflow.** These include: Insufficient power of the human, insufficient self-control/independence, insufficient reaction time, unrealistic expectations, delayed notification, insufficient support, and other process and workflow failures.
**3. Failure at the human–machine interface.** These include: Incomprehensible or incomplete outputs, complex or poorly designed user interface, insufficient training, and other epistemic failures.
**4. Failure of the human component.** These include: Cognitive bias, automation bias, fatigue, incongruous intentions, stress or overload, lacking courage, and other human-centric failures.
**5. Exogenous circumstances.** These include: Unreasonable laws, unreasonable societal expectations, inappropriate workplace requirements, and other external pressures.

C.1 gives supporting details on the empirical rationale and literature behind these categories, along with an extended table of failure modes. The categories are ordered by 'amount of human-ness', from purely digital failure (no human-ness), to failure of social pressure and wider society (vast human-ness). Our taxonomy covers HITL setups in general computational machines, of which ADMSs are an example. C.2 gives further discourse showing failure categories 2 and 4 are distinct.

## 3.2 CONNECTIONS TO OUR HITL SETUP TYPES

Setups configured for trivial monitoring, where the human cannot provide computationally meaningful input beyond 'proceed' or 'halt', may be particularly susceptible to failure modes related to the human component. The human's comparatively passive role means that automation bias, fatigue, or simply lacking the courage or perceived authority to halt a process become significant risks. This setup can also mask process and workflow design failures, such as the human having insufficient power to truly intervene or having unrealistic expectations about the level of oversight provided. Failures originating in the machine components themselves might also proceed unchecked, as the monitoring human may lack the mechanism or mandate for closer scrutiny.

An endpoint action setup relies on a single, critical human input after the machine has performed its computation, concentrating failure risks around that specific interaction point. Human–machine interface failures become critical: if the machine presents incomprehensible or incomplete outputs, or if the interface is poorly designed, the human may be unable to make informed decisions. Similarly, human component failures like cognitive bias can influence their single judgment with no backup checks present. Failures in process and workflow design, such as insufficient reaction time allowed for the human or delayed notification that input is required, are also highly relevant here.

In an involved interaction with potentially many queries, failure modes quickly become more complex. The back-and-forth nature of the process makes deeper design failures (e.g., unclear roles) and human–machine interface issues more pronounced. While increased interaction might help the human catch machine failures (and vice-versa), and break down a vast set of options into a manageable list of choices (see §2.3), it may also lead to unexpected machine behaviour if the machine adapts to the user in unintended ways. Human failures such as fatigue or stress from prolonged interaction, or the accumulation of cognitive biases across multiple decision points, can additionally degrade performance over time. Furthermore, the complexity involved can hide superficial engagement with the machine outputs when quantitative interaction is mistakenly understood as qualitative interaction: many smaller human–machine interactions may not sufficiently change the machine's output. Overall, while a HITL setup requiring involved interaction gives the human the most power to intervene, it is also the setup which is most complex, thus leading to more nuanced failure mechanisms.

In summary, different failure categories may be more likely for different HITL setups, and each category can be realised through different failure modes. We believe this taxonomy of five failure categories is able to capture *most* HITL failures. However, the list below each category should not be seen as exhaustive, but rather reflects a selection from the wider literature, with further discourse in C.1. They are there to illustrate that one should not only consider HITL setups via reductions, but also by simultaneously grouping different failure modes into our categories from §3.1. This provides a two-dimensional actionable picture of HITL setups focused on harm prevention. Akin to the oversight of general mathematical technologies (Chiodo & Müller, 2025; Müller et al., 2025), ignoring entire categories is a likely way to failure, while ignoring individual failure modes can still lead to failures in specific contexts. In short, reductions and failure categories should always be considered together.

## 3.3 EXAMPLES OF HITL SETUP FAILURES

This failure mode taxonomy enables us to identify where and why real HITL deployments fail. We can apply it to several case studies of (failed) HITL setups, showing how these setups failed in relation to the above taxonomy. What comes to light from these is that failures of HITL setups are usually due to poor integration (Müller et al., 2025), and what is often put down to 'technical failure' or 'human error' can actually be avoided if proper integration is carried out (as happened with an ADMS security scanning setup at a sports stadium, where two firearms were brought through security and the problem was blamed on 'human error'; see C.3 for further discourse). To further illustrate the failure facets of endpoint action HITL setups, C.4 gives further discourse on the catas-

trophic fire at Notre-Dame Cathedral. Here, we turn to a fatal self-driving car collision involving a trivial monitoring HITL setup (and C.5 gives further discourse on some mitigations, along the lines of our taxonomy from §3.1):

As presented in Hawkins (2018); Fitzsimmons (2018); NTSB (2019), in March 2018 the first recorded incident of a pedestrian fatality from a collision with a self-driving car occurred. The car had a trivial monitoring HITL setup created by Uber, with a human driver (an Uber employee) at the wheel poised to intervene and 'take over' the driving at any point if they observed a problem with the self-driving mechanism (NTSB, 2019, p. 8). In this HITL setup, the human operator monitored the ADMS responsible for autonomous driving and had the ability to intervene by braking and/or taking the wheel, but they did not have the ability to change the ADMS's decision-making in more complex ways. The human monitor was expected to perpetually 'hover their hands above the steering wheel and foot above the brake pedal' (ibid., p. 12) (unrealistic expectations), without ever touching them as that would disable the self-driving mode (ibid., p. 11). The car had been in self-driving mode for 19 minutes before the crash, with no tasks required of the human driver (ibid., p. 19) (fatigue, automation bias). The self-driving ADMS first identified the pedestrian 5.6 seconds before collision, as they were jaywalking across the road. However, as it had not been programmed to classify a jaywalker (ibid., p. 16), it spent the next 4.1 seconds misclassifying the pedestrian as 'another vehicle', 'bicycle', and 'other' (ibid., p. 15) (unexpected inputs). The ADMS first predicted a collision 1.5 seconds before impact and began evasive action; at 0.2 seconds before impact the ADMS established that impact was inevitable, instigated a controlled slowdown, and gave the human driver the first warning notification of upcoming impact (ibid., p. 16) (delayed notification, insufficient reaction time). The human driver had, from 6 seconds before impact, been gazing down at the control panel (ibid., p. 18), allegedly watching their cellphone (ibid., p. 24) (incongruous intention). With a warning lead-in time of 0.2 seconds—approximately human reaction time—the human driver had virtually no time to take control; they took the wheel 0.02 seconds before impact, and the car then hit the pedestrian. The entire HITL setup was affected by a poor safety culture (ibid., p. 38) (insufficient support, other external pressures). In this example, the failure cascade spans across our taxonomy: (1) the ADMS components failed by not recognising the pedestrian, (2) the workflow failed as the human allegedly was not embedded in a supporting safety-culture and had insufficient reaction time and unrealistic expectations, (3) the human–machine interface setup failed by allowing an environment whereby the human could watch a video, and (4) the human failed by being inattentive. However, one key further failure of this setup arose post-incident, when the law failed to hold the company accountable in any way or form (corresponding to (5) from our taxonomy), which brings us the final aspect of our manuscript: responsibility.

## 4 LEGAL–MORAL RESPONSIBILITY

We now narrow our focus to ADMSs. If a HITL setup fails and causes harm, legal and moral principles inform our understanding of what went wrong and who is at fault. The Uber case (§3.3) highlights the challenges when legal frameworks intersect with HITL failure modes. In this case, both criminal and civil liability fell on the Uber *employee*; the human operating the car. This was despite allegations that Uber's technology was flawed (i.e., not recognising the jaywalking pedestrian) and the company's poor safety culture (for case details, see Stamp (2024)). In the final section of this manuscript, as a 'proof of concept' we consider relevant (EU, UK) legal frameworks concerning design setups and liability (D.1 gives supporting details on why we chose these jurisdictions) and point to ways that they can be improved by incorporating our formalisation of HITL setups. We also identify a trade-off between responsibility and explainability in choosing setups. However, in all HITL setups where responsibility gaps emerge, we argue that there should be a more nuanced approach to liability informed by existing complex causation cases, to avoid the scapegoating of humans to shield technology companies, like in the Uber case.

### 4.1 HITL SETUPS AND THE LAW

The UK and EU General Data Protection Regulation (the *GDPR*, which only applies to *personal* data processing (Article 4(1) GDPR)) and the EU AI Act have imposed requirements for human oversight mechanisms to be implemented for automated decision making and *high-risk AI* (see Article 6 and Annex III of the EU AI Act). Both legal frameworks take a 'by design' approach, requiring

developers to embed certain safety mechanisms into the technical setup of their ADMSs before they are deployed. This means that the law does not only threaten legal liability when things go wrong (discussed later in §4.4), but it also mandates measures to *prevent* harm from occurring. Given the focus of the GDPR and EU AI Act in relation to HITL, lawmakers have acknowledged that HITL setups are a crucial tool to prevent harm in high-stakes scenarios like biometrics, law enforcement, and employment (EU AI Act, 2024, Annex III and GDPR, 2016, Article 22(1)).

Article 22(1) of the GDPR governs 'solely automated decisions', generally prohibiting trivial monitoring setups (with supporting details in D.2) that could cause legal effects (or similar) on individuals, like credit applications or e-recruiting practices (Vollmer, 2023, Recital 71, GDPR). To move beyond trivial monitoring and avoid falling within the scope of Article 22 of the GDPR (which requires consent and other safeguarding requirements), companies need to implement **'meaningful oversight'** (European Commission, 06/02/2018, p. 19): currently understood in this manuscript as an endpoint action. Similarly, the EU AI Act requires high-risk AI to be 'designed and developed in such a way, including with appropriate human–machine interface tools, so that they can be **effectively overseen**' by individuals while in use (EU AI Act, 2024, Article 14(1)). The wording 'meaningful' and 'effective' in both the GDPR and EU AI Act suggest that trivial monitoring alone is insufficient to meet legal obligations. A trivial monitoring setup with no influence on the decision or computational outcome cannot be doing anything 'meaningful' or 'effective'. Current laws focus on the role of humans at a very late stage in the HITL setup, in either trivial monitoring or endpoint action setups and as Sarra (2024, p. 4) notes, 'substantial human intervention in previous stages appears to be irrelevant'. But the law (and related guidelines) do not indicate what alternative HITL setups should be implemented (see D.2 for supporting details).

## 4.2 MOVING TOWARDS INVOLVED INTERACTION

Our computational classification of HITL setups provides a framework to compare the significance of the human's involvement, and their ability to reduce harms within each setup. More frequent human interventions have computational and ethical implications (§2.2, 3.2). We argue that stronger reductions involving at least some (and potentially unbounded) querying of the human should be required to implement 'meaningful' or 'effective' oversight as stipulated by the law. By setting out technical setups that align with legal requirements, developers are incentivised to design their ADMSs with greater human involvement, which can improve system safety. Beyond the oversight requirements, the EU AI Act and GDPR also prescribe certain safeguarding duties on the human, whereby they are expected to prevent or minimise risks to 'health', 'safety', 'legitimate interests', and 'fundamental rights'. Yet, in trivial monitoring or endpoint action, humans are not effectively enabled to perform their safeguarding duties because they may face a completely 'black box output' from a machine (§2.2). From such an output it could be impossible for the human to assess whether any rights or interests have been infringed, due to any inherent opaqueness of the ADMS's output and of the factors that influenced the decision.

By contrast, in involved interactions, the agency of the human within the sociotechnical system is enhanced and they become actively enabled to meet their safeguarding duties. In a HITL setup where the ADMS asks many questions, the human can better assess what should or should not be factored into the ADMS's output to make fair decisions that do not infringe rights or interests (§2.2). Thus, the human has increased agency to scrutinise the ADMS's process. Further, endpoint action or involved interaction setups enable the human to input their own information. Depending on what the ADMS asks, the human may be able to better align the output of the ADMS with human values, for example by confirming or denying the relevance of certain input factors, like religion, race, or sex. D.3 gives supporting details with an example of the benefits of stronger reductions (like involved interaction) with a recent legal case involving SCHUFA—whereby an automated credit scoring system with a (weak) endpoint action HITL setup was deemed by the courts as acting as a trivial monitoring setup (supporting details in D.2). As explained in D.3, by implementing an involved interaction (§2.1) to break the automation bias of the human (§3.1), SCHUFA could have prevented the 'slip back' to trivial monitoring and thus avoided violating Article 22 (§4.1). Of course, involved interactions are not a perfect solution as they still suffer all the potential HITL failure modes (§3.1, 3.2). But, in general, they enhance the agency of companies and humans, better enabling them to meet their legal and moral obligations to provide oversight as well as safeguard the decision subject's rights. But ultimately, without proper consideration for what actions the human can take *in theory* (§2), and the ways in which human–machine interactions fail *in practice* (§3), the human may have

to be *superhuman* to meet these safeguarding duties (§4.1). Specific learning strategies for operationalising HITL setups are given with supporting details in D.4, including learning to defer (L2D) and conformal predictions, and how they can facilitate involved interaction setups by triggering and aiding real queries to the human.

## 4.3  HITL RESPONSIBILITY AND EXPLAINABILITY TRADE-OFF

While an involved interaction may enable the human to positively impact and scrutinise the output of an ADMS, this comes with a trade-off which complicates legal and moral responsibility. 'Responsibility gaps' refer to situations where the 'black box' features of autonomous technologies, combined with the complexity of the sociotechnical system, obfuscate an immediate source of responsibility for the impact of an ADMS (assisted) decision (Matthias, 2004). Introducing (significant) human influence into a system has been posited as a way to reduce responsibility gaps when compared with systems with limited or no human influence (Sienknecht, 2024, p. 194). Indeed, using our reductions framework, in a HITL setup with a *weak* reduction such as an endpoint action or trivial monitoring, it may well be *easier* to directly link the impact of the ADMS with the actions of the human because they may have approved the ADMS's output, thus closing responsibility gaps. Contrastingly, in an ADMS with no human in the loop, 'the ADMS' is responsible (though assigning responsibility to an ADMS opens up a web of distributed responsibility, where a number of parties may share moral responsibility (Sienknecht, 2024)). But, as we now show, the link between responsibility gaps and HITL becomes far more complex as human influence increases further.

While we advocate for stronger setups (involved interactions) to meet GDPR and EU AI Act obligations, within these it is immensely complex to determine how the human input(s) impact the ADMS output due to the human–machine entanglement (see D.5 for supporting details). In §2.2 we described a chain of computation. But the longer this chain, the more obfuscated responsibility is, as it becomes harder to pinpoint which human and/or machine decision(s) had the most impact. While the human does have more impact on the ADMS, the impact of these inputs, even if recorded (supporting details in D.5), on the ADMS's output remains potentially unknown. By contrast, it could be relatively simple to identify the human's impact within a setup like trivial monitoring or endpoint action, because they effectively make a decision at the start whether to use an ADMS, and a decision at the end whether to use its output and if so, how. Here, the impact of the human on the overall ADMS and its output is clear, unlike in an involved interaction. As such, we witness a trade-off within involved interactions. On the one hand, they enable the recording of questions asked by the ADMS and the responses inputted by the human, increasing transparency and explainability of contributing factors (cf. §2.2). On the other hand, the human–machine entanglement erodes attribution of the human's impact on the ADMS's outputs, creating responsibility gaps from the complexity of identifying which decision point(s) led to system failure. This is an unavoidable explainability–responsibility trade-off: a more explainable HITL setup with clearer intermediate computational steps obfuscates responsibility; a setup with clearer attribution of responsibility is far more 'blackbox'. This trade-off needs consideration when using the law to motivate building better HITL setups, as the two can operate at cross purposes. In D.6 we give supporting details on how our legal and moral analysis applies to HITL setups built from the intermediate reduction types given in B.4, as well as a way to formalise HITL setups from those (and other) reduction types.

## 4.4  AVOIDING THE HITL 'SCAPEGOAT'

So far, we have identified omissions in the law's approach to moral responsibility gaps and the understanding of the technicalities of HITL setups. To avoid the human being treated as a scapegoat (like in the Uber case), we argue that regulations should provide guidance on HITL setups in terms of reduction types, to incentivise companies to design such setups more effectively. The GDPR and EU AI Act go some way to resolve these emerging responsibility gaps by imposing liability on the data controller or technology provider to ensure that ADMSs are *designed in ways* that enable 'meaningful' or 'effective' oversight to allow the human to safeguard the rights of decision subjects. This manuscript contributes by showing *how* this might be done from computational (§2) and practical (§3) perspectives. This is important because the onus is on the company, whether or not the human causes harm, to design setups effectively. Nevertheless, the scope of these legal frameworks is limited and claims may also arise in negligence law when harm has occurred. For example, the UK courts have previously departed from established principles in challenging cases

involving 'responsibility gaps' to compensate individuals who have been harmed but the cause of injury cannot be discerned (for supporting details see D.7). These cases involved workers developing deadly mesothelioma from exposure to asbestos fibres across multiple employers (House of Lords, 2006). There, liability was calculated by an amount relative to the proportion of exposure the claimant had at a given employer, even though they might not have actually caused harm directly. The principles underlying the court's treatment of the mesothelioma cases may provide inspiration for how we should address similar responsibility gaps in HITL setups with involved interaction (for further discourse see D.7).

In the US, one already sees a tendency to hold the human liable for the failure of an ADMS (Stamp, 2024). The authors are apprehensive about such liability approaches in the event of failure across all computational setups from §2.1.[1] It might seem like the most intuitive response, especially in trivial monitoring and endpoint action where the human often has final control over the actions. Yet, as our taxonomy in §3.1 shows, failures arise for multifaceted reasons which the human might have only limited control over. Even in involved interactions, where arguments from moral philosophy revealed that responsibility is more complex, the human should not be used as a complete 'scapegoat' to shield companies from accountability for their contributions to failures, even despite their increased agency in the sociotechnical system. The mesothelioma-style cases provide a foundation for how the courts should respond to this which aligns with the computational realities of involved interactions and the responsibility gaps that emerge at these intersections. Where the human also has onerous obligations to safeguard individuals and ADMSs are error prone, biased, and frequently act in unexpected ways, we need to avoid the human taking total liability for all failures.

## 5    CONCLUSION AND SUGGESTIONS

Our analysis brings new clarity to the design of HITL setups by characterising them through computational reductions and complementing this formalism with an empirically motivated taxonomy of HITL failure modes. HITL setups involve complex sociotechnical decisions and are susceptible to failures beyond human control, which necessitates this joint perspective for designing effective and responsible setups. Our analysis connected these setups and failure mechanisms to existing limitations and omissions in the law. This allowed us to make suggestions for more refined rules surrounding HITL requirements, as well as identify a trade-off between responsibility attribution and technical explainability, and recommend that courts cautiously approach liability in these cases. We thus make the following suggestions for those designing or regulating HITL setups:

1. Define the computational HITL setup type, if possible aiming for more than trivial monitoring.
2. Avoid 'bolting-on' HITL to existing workflows; it must be fully integrated into the process.
3. Establish guidelines for meaningful human oversight that consider different HITL setups.
4. Ensure that expectations placed on humans in HITL setups match their competency.
5. Implement measures to prevent humans becoming 'moral crumple zones' protecting machines.
6. Understand the trade-offs between causation and technical explainability, to inform more nuanced approaches to assigning legal liability in HITL failure cases.

If done poorly, a HITL setup can create a dangerous two-way deferral of responsibility between machines and humans. Humans may overly defer to machine computation, and machine designers may overly rely on humans for safety, all of which can lead to disastrous consequences. Integrating HITL is not a binary process; many ways exist, with different consequences. As a bad HITL setup may be just as harmful as no HITL setup, achieving a *good* setup needs to be the objective.

### AUTHOR CONTRIBUTIONS

The first author (Chiodo) conceived of and led the project. Authors 2-5 (Müller, Siewert, Wetherall, Yasmine) are listed alphabetically by surname, and all contributed equally. The final author (Burden) oversaw the project and provided high level direction. All authors contributed to drafting, writing, editing, and reviewing.

---

[1] Unless the human has acted in ways that can be proved as deliberately malicious or seriously negligent. But even then, ADMS designers are responsible for implementing controls that prevent human malevolence and create fail safes and checks on how the human interacts with the ADMS.

ACKNOWLEDGMENTS

John Burden acknowledges support from Effective Ventures Foundation—Long Term Future Fund Grant ID: a3rAJ000000017iYAA.

REPRODUCIBILITY STATEMENT

The vast majority of results and analysis in this manuscript are of a completely theoretical nature, and thus can be verified through further theoretical research. As outlined in C.1, the initial empirical foundation of the taxonomy of §3.1 is derived from a series of ethics consultations with different startups. These consultations were conducted between 2020 and 2022. As explained in C.1, we further substantiated the taxonomy by comparing our initial observations with the existing literature.

STATEMENT ON USE OF LARGE LANGUAGE MODELS

We have used Gemini Pro 2.5 Deep Research to help with literature search, and Gemini Pro to help with grammar and formulation in selected places.

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

# A    Introductory Material

The main purpose of this appendix section is to provide further details on some of the terminology we use that we did not have space for in the main part of the manuscript, which we now signpost here for convenience.

In A.1 we specify the interpretation of *harm* that we make use of in this manuscript, as well as the associated terms *risk* and *safety*, connecting these to existing literature. Next, A.2 further expands on the many terms related to HITL setups that appear in the literature, with references for each. And finally, in A.3 we discuss how existing conceptions of HITL are presented in the literature, how these relate to our work, and what we plan to do in the rest of the manuscript to extend and formalise these concepts.

## A.1    What we mean by harm, risk, and safety

Broadly speaking, we take 'harm' to mean the infliction of some form of damage, be it physical or otherwise. Related to this, 'risk' refers to the chance of some harm(s) occurring. And 'safety' is the reduction or mitigation of risk, and hence avoidance of harm.

We use these terms in the broadest possible sense when talking about the failures of HITL setups. For example, strictly speaking, the (human) driver of a car falling asleep at the wheel is not a 'harm', but rather a (normatively-defined) 'wrong' (Diberardino et al., 2024). However, it may rapidly lead to a (rather severe) harm, such as a car crash. So when we talk about harms, we also include these 'wrongs'; events and actions which, though not necessarily harms in their own right, would almost certainly lead to harms in the strict sense of the word. And the same goes for how 'harm' is interpreted in the definitions of risk and safety.

Substantial further work has studied the specific actions within sociotechnical systems that lead to material harm (Diberardino et al., 2024), as well as what these harms can potentially be (Shelby et al., 2023) and how one may take steps to actively avoid them (Dobbe, 2022). Indeed, it may be insightful to relate these notions to failure modes (§3) and HITL setup structure (§2) from this manuscript as part of future work.

## A.2    Existing terminology

The existing terminology describing human involvement with ADMSs varies with differing degrees of human interaction and control. HITL setups have humans actively integrated into the ADMS's operational cycle (here we exclude human input in the training or development phase), while Human-on-the-loop (HOTL) has humans primarily acting as supervisors who intervene only when necessary (Nothwang et al., 2016). Setups where the ADMS is designed to operate without direct human input or intervention can be termed Human-out-of-the-loop (HOOTL) (Wagner, 2011). A higher level perspective is given by Human-in-Command (HIC), whereby humans determine the high-level functioning of the ADMS (Anderson & Fort, 2022; Kowald et al., 2024). To bridge the legal and ethical responsibility gaps (Matthias, 2004) that can exist in such setups, the concept of Meaningful Human Control (MHC) was introduced to study how much influence a human should have on the execution of an action and what necessary levels of cognitive and moral awareness they should possess (Davidovic, 2023; Roff & Moyes, 2016; Abbink et al., 2024). In this context, Green (2022) speaks of three human oversight policies: 1) 'restricting solely automated decisions', 2) 'emphasising human discretion', 3) 'requiring meaningful human input.' Later, in §2, we show how existing HITL terminology can be understood from the perspective of formal computability theory, and use this to deconstruct (3) above into a more fine-grained categorisation.

## A.3    Existing conceptions of HITL

HITL setups can in certain situations mimic the trolley problem (i.e., the human needing to choose between multiple undesirable outcomes), particularly when the human has no ability to shut off the ADMS entirely, but often also go beyond it in legal, moral and technical complexity (cf. Steenson (2021)). Recent research suggests that while HITL setups lead to more uptake of ADMSs, they potentially also decrease accuracy (Sele & Chugunova, 2024). Green (2022) particularly discusses the empirical evidence for human oversight policies of ADMSs, arguing that these generally fail to

address the fundamental flaws in controversial government algorithms while simultaneously offering legitimisation of the algorithms and the protection of vendors and agencies from accountability. Elish (2019) similarly raises the point that HITL setups can end up protecting the ADMS rather than the human, speaking of humans as the 'moral crumple zone' taking on accountability and enabling potentially faulty ADMSs to stay in place. Overall, this literature suggests that a closer scrutiny of HITL setups and their failure modes is necessary.

The lack of a well-defined meaning of what a HITL setup can and does involve, and why such setups may be beneficial in sociotechnical systems, is a recognised problem. Recent literature has attempted to specify potentially desirable HITL setups in medical contexts. Salloch & Eriksen (2024) have argued that both clinicians and patients should be included as 'co-reasoners' in a medical ADMS context, making their own judgments about if, how, and why to use this technology, as well as how to use its results. Accordingly, they argue that such a HITL setup is valuable in that it forces questioning by both parties about whether to use ADMS technology, thereby justifying its use, reduces automation bias by encouraging doctors to not rely on an ADMS without justifying its use to the patient, and uses an ADMS not just as a tool to generate answers but as 'discussion' prompts based on the values and aims of both patients and doctors. Building off the setup in Salloch & Eriksen (2024), Griffen & Owens (2024) have argued for a kind of proliferation of HITL setups in medical ADMSs, highlighting the potential role of other clinical staff, and the value of having a diverse range of humans in these setups.

Recognising that human experts must play an active role in HITL setups, Natarajan et al. (2025) argue to shift the perspective and call such setups AI-in-the Loop (AI$^2$L), whereby it is the ADMS that is part of a larger sociotechnical (human-led) process. Recognising the different degrees to which humans are, and should be, involved in producing functioning HITL setups, we present a classification of these in §2.1 (using Turing and many-one reductions) to concretise the language for further legal and ethical analysis.

HITL is frequently presented as a critical safety measure in high-risk domains (EU AI Act, 2024), such as autonomous driving (Huang et al., 2024) and the military (van Diggelen et al., 2024). However, the scalability of human oversight has been increasingly questioned (cf. Chiodo et al. (2024)), especially for advanced ADMSs. The challenge of *scalable oversight* highlights fundamental limitations: human supervisors may struggle to meaningfully oversee ADMSs whose cognitive capabilities surpass their own across multiple domains (Amodei et al., 2016; Bowman et al., 2022). Our work in this manuscript, on both the formal classification of HITL setups, and their failure modes, will help shed light on the positive and negative safety aspects of different HITL setups.

## B    Computational Reductions

The main purpose of this appendix section is to elucidate some additional background and insights from reductions that we did not have space for in the main part of the manuscript, which we now signpost here for convenience.

In B.1 we give full details of what we mean by conceiving of the human in a HITL setup as an oracle. This includes a thorough treatment of the technical and formal machinery and background used, as well as certain clarifications on what actually happens in these setups in practice. Following on from this, B.2 gives a diagrammatic illustration of the three HITL setups we define.

B.3 covers how human agency motivates our HITL setup definitions. B.4 discusses intermediate setup types between endpoint action and involved interaction, and why we have chosen to avoid covering them explicitly in our analysis. B.5 gives an expanded description of our route-planning example demonstrating our three setup types.

We then proceed to cover the computational strengths (B.6) and socio-technical benefits (B.7) of involved interactions, and in B.8 give a diagrammatic representation of how they 'open the black box' of the computation for additional scrutiny.

We finish by discussing the difficulties of determining setup types in practice, and what additional approaches need to be carried out by developers to make determination more manageable (B.9). We give specific reference to setups which might appear to be one type, but in practice can be mimicked by a less-powerful one (B.10).

### B.1    Conceiving humans as oracles

In our formalisation of the notion of human-in-the-loop we represent the human by a fixed oracle function $f$ which is used in some oracle machine $T^\bullet$. This may suggest that we are assuming human decisions are transparent, deterministic, and truthful (perhaps even requiring some kind of omniscience). On the contrary, our view of human–machine setups as oracle machines implies quite the opposite.

The operation performed by the human in a HITL setup is precisely one which cannot be automated. Practically speaking, this is the reason the human enters the computation in the first place. The reasons for this difficulty in automation can be manifold, depending on the nature of the operation. It may be difficult for machines to do or even grasp, and involve questions whose answers are subjective, require life experience to give, rely on some sort of 'moral judgement', or depend on unforeseeable circumstances. Often the human may be confronted with questions that do not admit a 'correct' answer in the same way evaluating some arithmetic expression does, e.g., if they entail judgements on values, morality, or emotional response. In regards to subjectivity and morality, some fundamental computational problems are discussed in detail in Moor (1995), and more recently Bellaby (2021) argued that at least (ethical) decisions cannot be made by systems which are predictable. Hence these problems are left for the human to grapple with; a practicality whose outcome (almost by definition) cannot be predicted by the machine developer. If the developer did know what the human would answer on some query, or how the human might 'compute' such an answer, they could simply implement that response or computational process in the machine. This we formalise by saying that the developer implements the machine $T^\bullet$, which is essentially an algorithm that asks questions at certain times. Such questions are those which the developers cannot give an algorithmic answer for, and instead require a human to answer. The instance answering the questions is the oracle function $f$; the developer does not know the precise behaviour of such a function, but has to treat the outputs as computationally meaningful (see below). The human is to act as the oracle; they will answer the questions to the best of their ability (ideally), but of course may have a myriad of failure modes as discussed in §3.1.

Once the decision-making procedure is put into action, the human will provide some inputs that are processed by $T^\bullet$. Therefore, in effect, the human is providing some function $f$. Crucially, the developer has no influence on $f$ (or at best very weak influence) and hence the function is for all practical purposes fixed from their perspective—they have to design the system to make the best out of whatever the human will answer. Yet, this function need not be literally fixed in advance. The human could change their mind over the course of the decision-making process, and thus answer the

same question differently if asked later.[2] Indeed, it could be that in the morning the human flips a coin determining their moral compass for that day, or they are exhausted and behave differently than they would at other times,[3] or they have learned new things and improved their skills over time—many settings in which the human behaves in ways that are not transparent or even determined in advance are conceivable. It could even be that there are multiple humans who together provide the input, or different humans at different times or at different points in the human–machine interaction. Thus if we conceive of the human as an oracle, and say that any particular oracle is given by a function $f$, this does not impose any assumptions whatsoever on how humans behave. We also do not suggest in any way that the specific person to serve as the human in the HITL setup needs to be known in advance when we say that $f$ is for practical purposes to be viewed as fixed.

The distinction we make between the properties of the oracle function and the computational power entailed by the oracle machine mirrors that encountered with random machines.[4] In a random machine, any particular computation is guided by an oracle function $f$ which is a bona fide (deterministic) function. However, the point of random machines is that the oracle function is drawn from a probability distribution which then induces a probability distribution on machines (hence, computed functions). In this sense, 'the oracle' is random, because the probability distribution on $f$ is what introduces randomness into the computation. This is even though any particular oracle *function $f$* itself is deterministic, and some of these may produce the correct output while others do not. A random machine obtains its strength through the fact that it is designed to call a random oracle, for which the properties of any individual oracle function are irrelevant (indeed, $f$ could be *any* function from words to words); it only matters that *most* oracle functions yield the correct output. Likewise, a decision-making procedure involving a HITL setup obtains its strength through sufficient use of the human, but it does so precisely because the behaviour of the human is not predictable. What is important for us is not so much what function $f$ is 'computed' by a human oracle, but rather how any reasonable such function could fit into the decision-making procedure modelled by $T^\bullet$.

Along the same lines, we would like to make a few clarifications.

1. In the computation, the machine $T^\bullet$ has to use whatever answer the human provides, treating it as 'true' to some extent. This is a trivial observation: if the human input was treated as completely untrustworthy, it could not be used to generate any insight. Thus a developer has to allow the oracle to answer questions 'however it sees fit' and design the system to process any reasonable answer as 'serious'. This of course does not mean that the (human) oracle is actually truthful or 'correct', or is even providing the 'best possible' response. Indeed, developers should take into account that it may not be.

2. As we point out in §4, there may also be legal or moral requirements for a human to evaluate and respond to a question arising in the decision-making process, even if it seems a machine could answer it. In this case, the contribution of the human is moral, rather than strictly computational, as phrased above. Still, the human plays an integral role if the overall decision by the human–machine system needs to reach some legal or moral status, and so the human is needed to produce the output.

3. We do not mean to imply that a literal Turing machine which operates in the way we describe would be a *practical* model of modern computational systems. Rather, our perspective is that it seems productive to transfer notions from computability theory to human–machine systems. Our formalisation is supposed to enable this conceptual move.

To summarise, the point of describing HITL setups in terms of the formalism of oracle machines is not to view (or idealise) humans as oracle functions, but rather to view human–machine decision-making setups as oracle machines.

---

[2]This can be modelled by the function if the time at which the query is asked is part of the 'input' the human receives.

[3]To give just one example from a vast literature on physiological effects on decision-making: radiologists' judgements on prostrate imaging results become more pessimistic later in the day (Becker et al., 2024).

[4]See van Melkebeek (2000, p. 33) for some introductory discussion of random Turing machines; there the oracle is just a string which is read bit by bit, but this does not affect the argument.

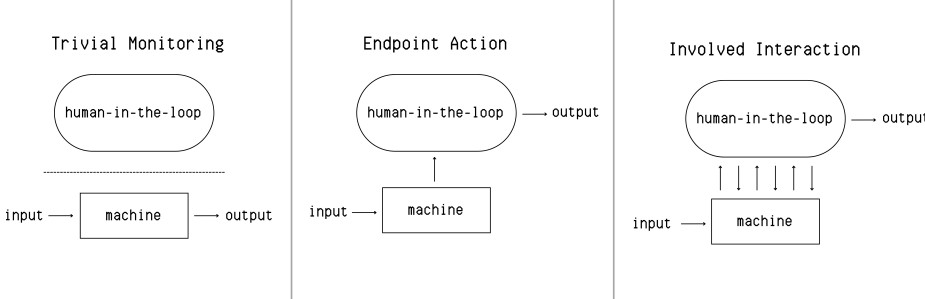

Figure 1: Operational diagram of each of our HITL setups.

## B.2 DIAGRAMS OF HOW OUR HITL SETUPS OPERATE

In Figure 1 we illustrate the operation of our HITL setups: trivial monitoring, endpoint action, and involved interaction. Note the computational 'ping-pong' between the machine and the human in the involved interaction setup, which significantly increases human input into the process.

## B.3 THE MOTIVATION FROM HUMAN AGENCY

The value of human agency in the decision-making process informs our definition of endpoint action and involved interaction in a subtle way. Recall that while computationally, any many-one reduction is a Turing reduction, a setup designed for endpoint action is by definition *not* an involved interaction. This is because the strength of an involved interaction comes precisely from the fact that it is not 'just' an endpoint action, as discussed in B.6. And the definition of involved interaction not only excludes endpoint action in this sense; it must also never have the option of behaving as an endpoint action. To clarify, consider an ADMS which has two possible 'modes' depending on the input: either it makes a decision and presents it as an output for a human to review, or it enters a multi-stage collaborative process with the human that allows for substantial contribution from the human. This would be an instance of an endpoint action, because the machine can 'choose' whether to involve the human. While such systems have many practical advantages, the authors believe that they do not give the human the agency associated with an involved interaction. Likewise, a computational process which can terminate with no contribution from the human at all is trivial monitoring, regardless of how the process behaves in other scenarios or even on average. We explain our rationale for this aspect in our definitions in detail in D.4; the *option* of agency does not actually imply agency.

## B.4 INTERMEDIATE HITL SETUPS

Why have we omitted discussion of restricted cases of involved interactions, such as where the machine can only ask the human oracle a bounded finite number of real queries (corresponding to a Turing reduction where the oracle can be consulted only a bounded number of times)? Computationally, this would lie properly between a many-one reduction and a 'full' Turing reduction. This omission is deliberate, as it would invite discussion also of the many other intermediate reduction types between many-one and Turing.[5] Conceptually, an endpoint action describes the 'simplest' way a human can interact with a machine in a computationally meaningful way, and an involved interaction describes the 'most complicated';[6] these two types alone give us a deep and rich area to analyse. It may well prove valuable to replicate the technical, legal, and moral analysis done in this manuscript on some of these intermediate reductions (e.g., when the machine can do its calculation, hand over to the human to do some more, then take back the human output and do some final machine calculation; having 'one round of ping-pong'). The technical, legal, and moral possibilities

---

[5]Including bounded truth-table reductions, truth-table reductions, and weak truth-table / bounded Turing reductions (the last of these we discuss in further detail in B.10). See Soare (1987, p. 83) or van Melkebeek (2000, Section 2.4) for details.

[6]Moreover, these intermediate reductions are strictly nested. That is, many-one reduction $\Rightarrow$ bounded truth-table reduction $\Rightarrow$ truth-table reduction $\Rightarrow$ weak truth-table / bounded Turing reduction $\Rightarrow$ Turing reduction (Soare, 1987, p. 83). In each case, there is progressively more use of the oracle.

may well be endless, so we have elected to examine the extremal cases for now: endpoint actions, and involved interactions, which correspond to many-one reductions and Turing reductions respectively. However, in D.6 we return to these intermediate reduction types, to see what of our analysis in this manuscript can be carried over to HITL setup types based on these intermediate reductions. There, we provide an avenue for future work by giving a generalised way to convert that family of reductions (or indeed any family, under certain conditions) into a family of HITL setup types.

The key notion for our purposes is thus that of the real query. Some care is needed when applying this notion in practice, for both technical and moral reasons. To illustrate this very simply, consider a machine that attempts to guess an integer $n$, where it is known that $n = m + k$ for some integers $m, k$ about which nothing is known. The machine might ask whether $m$ is even and then whether $k$ is even. According to our definition, the first question is not a real query because by itself it does not change the set of possible outputs (any integer can be written as an even integer $m$ plus some integer $k$). The second question then is a real query, as it determines for instance the parity of $n$. However, if the first question was somehow omitted, the second question would not be a real query for the same reason the first is not. This simple example shows that whether something is a real query depends not just on the question but on the whole computation. These interactions between questions and their implications on the output may be complicated in some systems. The key point however is simple: a query is only substantial insofar it leads to a real query.

A related question which arises from our analogy to computability theory is how resource-limitations on the machines and oracles in question play a role. In the theory of computation quite substantial attention has been given to the question of what happens if oracle machines which run in polynomial time are given oracles for various hard but computationally possible problems. Such is the theory of complexity classes from and around the polynomial hierarchy, including P, NP, BPP, $\Pi_2^P$, and PSpace (see van Melkebeek (2000, section 2) for definitions of these classes and (ibid., Section 2.4.3) for complete problems). Mathematically this theory is quite different from the theory of reductions without resource-limitations (i.e., classical recursion theory). To some extent our taxonomy of failure modes (§3.1) might be viewed as a comment on the resource limitations of human oracles. It may be useful to explore such viewpoints in later work, but we refrain from doing so as it does not seem to clarify the main points of this manuscript.

### B.5 ROUTE PLANNING MACHINES DEMONSTRATING DIFFERENT HITL SETUPS

Consider a route-planning machine in a HITL setup with the human driver of a car. It may demonstrate any one of the following HITL setups:

*Trivial monitoring:* The human could enter the origin and destination, and the machine could then present a driving route. The human then has the choice to take the route, or not. Here, the human has no input to the computation process, and can only 'turn off' the machine (by ignoring its output).

*Endpoint action:* The human could enter the origin and destination, and the machine could present the human with several different options, perhaps labelling them as 'fastest', 'most fuel efficient', 'most reliable', 'passes fuel stations regularly'. The human can then choose between these. Here, the human takes over at the end of the machine's work to finish the computation and produce the route to be taken (from the list presented to them).

*Involved interaction:* The human could instead input into the machine 'I want to visit my sibling'. The machine could then start computing, and asking the human a series of questions. It might start with 'when are you travelling?', take the answer, and do some more computation before coming back with 'when do you need to be there by?', then take that answer and do a bit more computation and come back with 'what do you need to bring?', then 'is anyone else travelling with you?', and so on, each time doing some computation between each question. These questions cannot be stacked all at the start, as the answer to earlier questions might determine which later questions are asked (and the number of questions may not be a priori bounded at the onset). The machine *may* then present some sort of optimised driving route. Or it may produce a very different output, such as suggesting to take the train as there is bad traffic, or to go on another day closer to the sibling's birthday. Or it may even advise against travel due to adverse weather. Here, the human has regular, meaningful, and not a priori bounded input into the computation.

## B.6 THE STRENGTH OF INVOLVED INTERACTIONS

Computationally, a Turing reduction between functions is viewed as weaker than a many-one reduction, as many-one implies Turing. This comes from a viewpoint that *fixes a computational problem*, and then asks what the space of oracles it reduces to is. So a Turing reduction is considered weak, because the problem reduces to more oracles; a many-one reduction would reduce it to fewer oracles so the reduction itself is strong. Thus, if one tries to many-one reduce a given problem to some oracle rather than Turing-reduce it, then one (generally) requires a stronger oracle, which in the context of HITL setups means a more competent human. Therefore, our concern is the opposite: we advocate maximising the space of functions that can be computed with a *fixed oracle* (a human). With that perspective, a Turing reduction is considered strong, because more problems Turing-reduce to the given oracle than if we used many-one reductions instead. In a HITL setup, the oracle is fixed (the human), so limiting the setups to endpoint actions limits the space of problems that can be solved. Therefore, making the space of problems solvable as large as possible is achieved by maximising the influence of the human, i.e., by using an involved interaction setup.

## B.7 THE BENEFITS OF USING INVOLVED INTERACTIONS

What exactly are the benefits of an involved interaction HITL setup? In short: because human (oracle) influence can add valuable, desirable insights throughout the overall computation, in particular at times where it is most needed by the machine.

Firstly, the human has more influence on the overall computational process, and thus on the outcome. With more, potentially unbounded, real human queries, the *agency* of the human is increased. Secondly, and related to this, the human has more opportunity to input their judgements and values into the overall computational process, thus giving more opportunity for the human to ensure the machine computation is *aligned* with its values. Thirdly, by intervening more often, the human has more opportunity to identify (and rectify) problems and safety issues within the machine computational process, before a final output or behaviour occurs. Increased human queries 'bakes in' the potential for increased human scrutiny. This aids with the *safety* of the HITL setup. Finally, and related to this, with more human queries, the machine is articulating intermediate steps that are human-interpretable more often (as a human input is sought), thus aiding with *transparency* of the computational process.

Overall, these four aspects come together to improve the *reliability* of the HITL setup.

## B.8 OPENING THE BLACK BOX VIA INVOLVED INTERACTIONS

In Figure 2 we illustrate the gradual 'unmasking of the black box' as one increases human involvement, from trivial monitoring, to endpoint action, and involved interaction. Note the chain of computation in the illustration of the involved interaction setup; this is a different way to view the computational 'ping-pong' between the machine and the human, and shows how this ping-pong serves to help unmask the black-box computation by the repeated handovers between the machine and the human. The human activity, and human–machine handovers, are all very interpretable to an observer.

This is because each time the machine needs to 'ask a question' to the human, it must precipitate its inner computation in a human-interpretable way, and then take a (human-interpretable) input. As more such questions are asked, the overall box becomes proportionately more filled with these human interpretable questions-answers, and so the black-box steps make up less of the process. As the number of such questions becomes very large, the remaining 'black box' parts of the computation become proportionately quite small, allowing humans to get a very good overall understanding of the computation (even if they cannot see every minute detail). As shown in Figure 2, the trivial monitoring setup is completely black-box, the endpoint action setup has a large black box process (but significant human-interpretable process), and the involved interaction setup is mostly human-interpretable steps (with very small black-box processes between them).

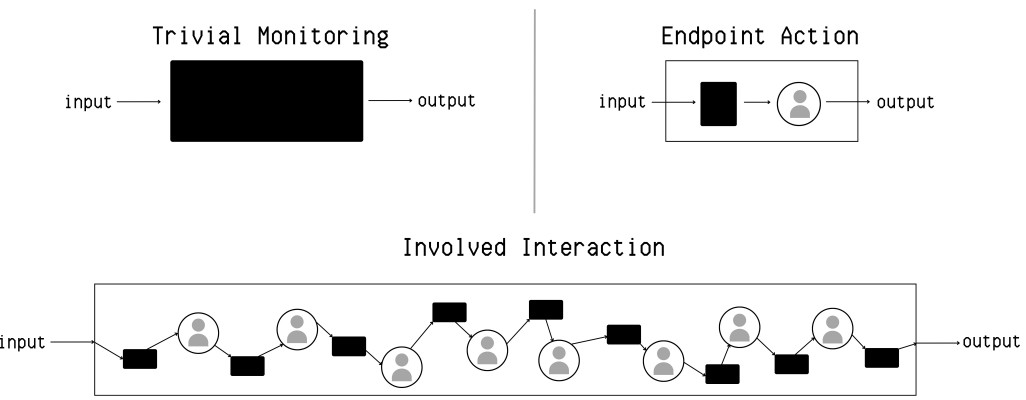

Figure 2: How real queries in HITL setups can unmask black-box computation.

### B.9 DETERMINING, TESTING AND DOCUMENTING HITL TYPES

The formal description of HITL setups presented in §2.1 is an important aspect of our manuscript. It is heavily relied upon later in §4, giving ways to prevent developers from circumventing the regulatory *intentions* behind any informal definitions of what a HITL setup should be. In particular, our definition of a real query, and the associated formalism around how each HITL setup type makes use of real queries, prevents such setups from having a 'veneer' of significant human involvement. The formalism describes these setups in a way that at the very least makes it hard to 'cheat' by implementing them 'in name only' without the meaningful effect intended. However, determining HITL setup types, and what they achieve, is a challenging task from each of a technical, legal, and moral perspective.

From a technical perspective, showing the non-existence of a reduction is difficult. While it is fairly straightforward to demonstrate some setup constitutes, say, a many-one reduction, by simply inspecting the structure of the reduction, that would not (in our terminology) show that a HITL setup is an endpoint action. To achieve that, one would also need to show that the machine always asks a real human query during the computation. Similarly, to show a HITL setup is an involved interaction, one would need to show that it is a Turing reduction, but also show that the machine always asks potentially unbounded and more than one real human query during the computation. Heuristically, while it is 'easy' to show that there is a human in the loop (as this is simply done by saying that a human is present), it is much harder to (computationally) verify how much the human will actually be asked to do.

From a legal perspective, showing that the human will do something 'meaningful' is also difficult (as discussed in relation to the SCHUFA case in §4.2 and D.2). While one could have a HITL setup where there are real human queries which are completely deterministic and replicable by the machine (e.g., asking the human 'What is 1+1?'), asking the human a series of 'pointless' questions (that the machine could just as easily carry out itself) may violate the legal principles of 'meaningful' or 'effective' human input, as discussed in §4.

From a moral perspective, showing the HITL setup is not simply a facade masking a less-involved process is also difficult. One could have a HITL setup where there are human queries which the machine cannot do itself (i.e., bringing in genuine human judgement), but then the remainder of the (machine) computation simply ignores the input. This would be an instance of a crucial query that is implemented in a way that is not 'real' in our sense. Even if the query is real (computationally), it might be that its impact on the computation is very small, or worse still, the machine might actively invert or go against the input. In reality, whether a query allows the human to meaningfully influence the outcome is not described by the binary notion of real queries. In any case, asking the human a question with a (seemingly) important answer, and then not using that or inverting the answer, would at best be a terrible oversight by the machine developers, and at worst be completely disingenuous and in opposition to the aim of HITL strategies. If not properly justified, and articulated to the human

in the loop and to the individuals affected by the decision, it may not only undermine the human's agency (a problem also discussed in the context of learning to defer in D.4) but also undermine their expectations about how a moral socio-technical system like a HITL setup should be functioning. To follow the spirit of our HITL formulation, such a *decision to ignore* would still need to be subjected to human oversight, albeit potentially not from the human whose input was ignored. If the machine were to actively go against a human input, with no human option to check or reverse this decision, and that had been a deliberate design feature, then that would be completely immoral from the perspective proposed in this manuscript.

However, some of the difficulties in verifying a HITL setup type from any (or all) of the three perspectives as described above implicitly assume the determination is done by some external actor (say, an auditor, or regulator, or colleague from a separate 'compliance' department), on whom all the onus sits to make such a determination. The implication above is that such an actor is given the ADMS to 'check', on their own, and so all the difficulties above arise. But that need not be where the full 'burden of proof' has to lie. Instead, one can shift some of this burden of proof to the *developers*, and require them to *demonstrate* particular behaviour or structural aspects of the system. Mechanisms for this already partially exist, or can be extended, in the form of required documentation, formal verification of certain aspects of the software, testing/benchmark regimes, etc. As we now show, in such an environment, the difficulties mentioned above start to become more manageable.

For example, model cards for model reporting as developed by Mitchell et al. (2019) already include documenting the intended use and ethical considerations. These sections could be (substantially) extended to include documentation of the HITL type and how the developers and integrators of the ADMS have attempted to mitigate certain failure modes. In combination with learning to defer and other learning strategies to operationalise involved interaction setups (further discussed in D.4), data sheets for data sets (as developed by Gebru et al. (2021)) could also play a critical rule if they are (substantially) extended to include a discussion about the relative strengths/weaknesses of the training data in relation to the human(s). Crucially, any standardised documentation of the HITL setup/type must focus on the integrated socio-technical nature of the entire setup; it is not enough to only consider the data, model, or human independently. We note, however, that such approaches require extensive further research, and the development of potentially new formats. As the formalisation presented in this manuscript is novel, it does not fit neatly into existing documentation approaches without (potentially substantial) changes and additions to them.

Salgado-Criado (2025) argue that the culture within the organisation developing the AI does not just influence how socio-technical setups for AI oversight are developed, but also how they are evaluated by the organisation. Critically, any form of testing and documentation thus requires educating developers about the subtle differences between human oversight and human control as they pertain to HITL setups. Proper verification, testing, documentation, and adjustment of the HITL type requires developers with an awareness of *ex ante*,[7] real-time, and *ex post*[8] aspects of control and oversight. Manheim & Homewood (2025) have developed a five-stage 'AI supervision maturity model' to measure and document how developers engage with these aspects, ranging from level 1 (a team does not engage with control and oversight in a well-defined and structured way) to level 5 (a team has properly identified risks, implemented measures, and communicated these to all relevant parties).

Writing about the development and usage of AI in medicine and nursing, Scott (2025, p. 1) argues that '[s]imilar to the frontier days of rapid expansion and lawlessness, AI is aptly linked to this time-period. It is moving fast and at least in the United States, there are unregulated, unchecked, and unethical applications to its use.' The above difficulties are certainly present in a regulatory and cultural environment whereby few restrictions on new technologies and requirements for documentation exist, while 'checkers'[9] have to check it all from scratch. But if the technicalities of the system, and its development processes, must be designed and presented in a way that demonstrates certain behaviour of the system, then one is no longer dealing with an 'arbitrary piece of code'. If a

---

[7]Meaning 'before the event'.

[8]Meaning 'after the event'.

[9]A 'checker' could include any of the following: auditor, regulator, developer, consultant, etc. And such people may be internal, or external, to the organisation creating the ADMS. We wish to consider any and all such people or entities tasked with carrying out such checks.

checker is instead tasked with checking a particular verification/proof, or documentation of a development process, holds, that is a much more tractable undertaking. Under a regulatory environment such as this, one might therefore consider the following approaches to determining a HITL setup type:

From a technical perspective, developers might be able to provide formal (mathematical) proofs to show that their setup is genuinely of a given type. However, further research is necessary into the exact nature of what constitutes such a proof, and when exactly it is possible to provide it. Inspiration may be taken from the extensive literature on formal software verification, but as we wish to make clear throughout the remainder of this appendix it also requires substantial further research and changes in the development process.

From a development perspective, formal verification necessarily means designing the system in such a way that this can be formally verified: a HITL setup may unlikely be verifiable if such aims were not considered during the initial design and development phases. Such a shift necessarily restricts the scope of what can be built, but at the same time adds the ability to demonstrate what the system does. In particular, developers would need to show that the setup does ask real queries, by showing the human will be consulted (some number of times), and that different responses lead to different outputs by the machine. It is already often the case that checkers are required to check for system properties that are incomputable *in general*, as this applies even to the Halting Problem. So by limiting what is designed and how, proofs can be provided which can then be checked.

Regarding any formal approach, however, it is important to consider that not all aspects from traditional software testing necessarily translate to all ADMSs. For example, Aleti (2023) note that testing LLMs introduces an 'oracle problem' (whereby LLMs often produce subjective outputs that cannot be measured against one objective standard) and problems regarding the comprehensiveness of tests (the tasks given to LLMs are often diverse, so a single test suite is unlikely to capture everything). Both can have a significant impact on verifying that an LLM-based ADMS setup constitutes a specific HITL setup. In particular, any requirement to formally verify a HITL setup type may result in different restrictions for different types of ADMSs.

An additional difficulty in measuring the effectiveness of a HITL setup comes from having limited data about the human(s) involved and how they behave (both independently, and while interacting with the ADMS). To learn more about how humans react to and use specific AI systems, and how that may affect HITL setups, Chen et al. (2022) have suggested a crowd-based methodology (HINT: Human-AI INtegration Testing). Their method is designed to gather information about how humans interact with (and oversee) a specific ADMS. While such large scale sampling is not always possible, the information gained from such sampling can be particularly helpful if multiple humans are involved or if their characteristics and behavioural traits are not yet known to the developers during the development of the ADMS. Such methods may, in certain circumstances, allow developers to not just formally prove the existence of an involved interaction setup, but also to empirically test for specific failure modes. We note, however, that substantial research is required into the dynamic nature of an involved interaction, and what this means for the consequences of an *existence* proof for such a HITL setup. For example, how does *knowing with proof* that a certain HITL setup constitutes an involved interaction change how the human functions in the loop?

From a legal perspective, developers might demonstrate what sort of questions the ADMS will ask the human, at what stages in the computation this will occur, and how such inputs will be used, with full documentation. In relation to Article 14 of the EU AI Act, Langer et al. (2025a) argue that this can range from a checklist-based approach to empirical testing, whereby the former focuses on straightforward checks such as checking certain best practices (e.g., Has the human been trained on specific failure modes? Is there an actual way for the human to stop the ADMS? What mechanism is used to force the ADMS to ask a question?), and the latter validates that the socio-technical system works in practice (e.g., Can the human actually perform their oversight function effectively, at all times, including during high-volume, fast-paced, and other high-stress events?). Langer et al. (2025b) see a necessary trade-off between efficiency of documentation and testing, and effectively mitigating risks. The problem of holistically testing a HITL setup often appears practically intractable: they note that proper (real-world) testing and documentation requires substantial resources and expertise that goes beyond specific technical knowledge, including psychology, law, and human-computer interaction. Additionally, they argue that all actions and insights are difficult to scale, as both the testing and documentation is highly tailored to one specific HITL setup; if the

ADMS, the human, or the application domain changes, some (or all) aspects of testing and documentation have to be done afresh. Whilst we do suggest that shifting (some of) the burden of proof onto developers will solve some of these problems, doing so requires further research to turn the open problems that Aleti (2023) and Langer et al. (2025a) identify into something that can be done in practice. Nonetheless, we believe that progress can (eventually) be made on these, at least in some settings.

More concretely, this requires extending existing, and developing new, handbooks and regulatory guidelines for developers, articulating what needs to be achieved. The developers might also provide active demonstrations for testing, and face 'reverse checks' where checkers ask what human inputs would be needed to achieve a particular outcome. Such reverse checks could be understood to be the human equivalent of counterfactual explanations for the ADMS decision (cf. Molnar (2020)). Testing, and documenting, whether that outcome is achievable by fixing some of the human inputs in a certain way may need to be part of it: the previous section on the technical perspective discussed how to demonstrate the presence of *real* queries; here we now seek a demonstration of whether such queries are *meaningful* by seeing if certain outcomes are actually achievable.[10] While it remains difficult to determine what the 'minimum requirements' for legal compliance would be with respect to the UK/EU GDPR and EU AI Act, since it will also be contextual, 'good practice' regulatory guidance can assist developers to meet their responsibilities. A further example of this is the 'Explaining decisions made with AI' regulatory guidance from the Information Commissioner's Office and Alan Turing Institute (2022). Part 2 of the guidance is aimed at technical teams and includes guidance on explainability and interpretability requirements for ADMSs. Going further than this, we suggest that a 'checker' can then examine everything produced by the developers along these lines (demonstrations, documentation, etc.), and observe whether any relevant legal requirements are being satisfied.

From a moral perspective, Kemell et al. (2024, p. 13) crucially note that '[e]thics is not just numbers' when it comes to machine learning operations, meaning assessing them has to go beyond measuring and verifying their performance. Still, from a moral perspective, there are certain technical aspects that can be considered to paint a picture of the functioning of a HITL setup. For example, developers might create recording mechanisms to log precisely what questions the ADMS asked the human and what answers were given. This can then be retained, consulted, and presented at the end of the computation, either for checking by the human in the loop to see if the 'spirit' of their answers had been preserved (perhaps necessitating the setup *ending* on a real human query in this way), or by an external checker to see if this had occurred (done at the time, or in a testing environment, or long after the fact). One could then see whether the ADMS is systematically ignoring (or worse still, inverting) the human responses by comparing the final output to what those responses were, to see if the output has remained 'morally aligned' with the human inputs given. While such logs do not turn a black box into a white box, a sophisticated logging mechanism not only helps developers understand the HITL type at the present time, but also helps them improve the ADMS over time, and helps the human(s) to learn about the ADMS and about their own strengths/weaknesses, enabling them to make their own necessary adjustments over time.

It is important to note that developers of ADMSs may face substantial hurdles, and competing interests, when they attempt to log interactions and document the HITL setup's moral shortcomings. For example, Müller et al. (2025) argue in the context of resort-to-force decision-making that power imbalances between developers, integrators, higher-level decision makers, and users can lead to decisions being overwritten when moral insights clash with what upper-level decision makers perceive as pressing economic and political realities. Just as Ho-Dac & Martinez (2024) see the need for international standardisation to eliminate technical barriers in oversight, there is a need for standardisation of testing and documenting moral aspects of HITL setup types: while the concrete morality of HITL setups necessarily differs and is context dependent, many of the questions that need to be asked, and documented, have a more universal character to assess whether the HITL setup embodies the positive principles of HITL design regarding human agency, value alignment, and non-scapegoating (as further discussed in §4). This character is similar to the universality of questions, and context-dependency of answers, for model cards and data sheets, and to the universality of questions observed for more general mathematical work by Chiodo & Müller (2025). Universal

---

[10]A mortgage determination tool may ask a series of real queries, but if postcode is fixed at a certain value then it may come to pass that the set of possible lending amounts are all sums of money under $1000; hardly a meaningful distinction between those and a mortgage rejection.

requirements and best-practices can help developers to achieve the difficult task of verifying, documenting, and maintaining a specific HITL setup type, and to defend it in potentially adversarial circumstances.

In the context of cryptographic systems, Bellare (1999) discusses the idea of 'practice-oriented provable-security' that bridges theory and practice; it is not enough to prove that a theoretical protocol is secure, one also needs to show that the resulting protocol implementation is practical, as if it isn't then that renders it unusable and thus hampers overall security. We also see theory-practice tension regarding the verification, and documentation, of HITL setups: while the formalisation presented in this manuscript suggests that something akin to 'provable HITL setup types' may be possible, the failure modes, moral and legal complexity, and general difficulties associated with machine learning operations and machine-computer interactions, suggest specific trade-offs are almost always necessary when attempting to verify, test, and document HITL types. The rather technical nature of 'practice-oriented provable-security' turns into a fully *socio*-technical problem for HITL. This associated complexity is also seen by others. For example, Kioskli et al. (2025) develop 'trust-Sense': a questionnaire-based tool focused on measuring the 'maturity' of an oversight approach, and thus, in particular, the 'maturity' of the development team and organisation. Shifting some of the burden of proof from (external) checkers to developers requires this kind of holistic look at the risks and mitigating measures taken during the entire lifecycle of a HITL setup, so inspiration may be taken from such approaches.

And so, whilst we envisage various technical *tools* to assist both developers and checkers in the demonstration and verification of HITL setup type, we neither imagine, nor advocate for, any fully *automated* processes to do so. The methods described above all involve substantial human participation, in accordance with our argument throughout this manuscript that better outcomes can be achieved with human-machine interplay rather than either on their own. Overall, by shifting some of the burden of proof to the developers, the role of any checker (the things they can check for, and how effectively they can check them) evolves from a completely unenviable one fraught with numerous difficulties, to one that appears more feasible. The difficulties in determining HITL setup type given at the start of this appendix begin to look more manageable when the checker is not left entirely on their own, but rather actively assisted by a regulatory and cultural environment that places requirements on developers to design, develop, document, and demonstrate aspects of their ADMS and associated HITL setup.

### B.10  BOUNDED TRUTH TABLE REDUCTION

In the endpoint action scenario of our route-planning example in §2.1 and B.5, the machine could have instead asked the human a fixed finite series of questions such as 'Do you prioritise speed or efficiency?', 'What is your maximum acceptable distance between fuel stations?', etc., and done computation between each, to find a single route to then present. This is an involved interaction, but in a precise sense it is not 'fully using the potential' of a Turing reduction. Namely, this process corresponds to a type of intermediate reduction as mentioned in B.4: a bounded truth table reduction. The crucial difference is that the uses of the oracle tape are bounded independent of the oracle. This process can be viewed as one that presents only one query and performs only a trivial operation after this. Namely, an oracle machine that asks a (human) oracle a (predetermined, finite) set of questions 'in series' one after the other, is computationally equivalent to one which does a slightly different computation with these options 'in parallel' and presents the conglomeration of questions to the oracle at the very end, phrased as one (big) question. So essentially these setups are the same as those which have only one real query, which are slightly stronger[11] than those for endpoint action, but weaker than those in involved interaction. Not all Turing reductions are made equal, and even among those which are not many-one reductions, some may behave differently than others from the perspective we advocate here.

---

[11]The Halting Problem does not many-one reduce to its complement, but there is a Turing reduction with only one query. It remains to be discussed when this distinction matters for human oracles.

## C HITL FAILURE MODES

The main purpose of this appendix section is to motivate and expand upon our taxonomy of HITL failure modes, with more detail than would otherwise fit in the main part of the manuscript. We now signpost this here for convenience.

The main part of the appendix, C.1, explains the origins and motivation of our taxonomy of failure modes, based in part on our experiences in industry. This is supplemented by extensive existing literature on the topic which supports our taxonomy. We also include a much-elaborated table of failure modes for our five failure categories. In C.2 we clarify, by way of examples, how two similar-looking failure categories are genuinely distinct.

To round off the appendix we give two case studies, covering additional examples of failed HITL setups that have occurred. The first of these (C.3) is covered in general terms. The second (C.4) is dissected and analysed according to our taxonomy from §3.1, as was done for the case study in §3.3. We finish in C.5 with a description of how our taxonomy could have been applied to prevent the failure and subsequent harm of the HITL setup in our case study in §3.3.

### C.1 LITERATURE ON HITL FAILURE MODES

The empirical foundation of the taxonomy given in §3.1 is primarily derived from a series of ethics consultations conducted between 2020 and 2022. During this period, one of the authors served on the ethics advisory board for the AI startup accelerator program *Machine Intelligence Garage*, which was part of Digital Catapult UK. As part of this engagement, the author provided ethics consultations to over 25 AI startups and conducted intensive, in-depth ethics workshops with three additional companies.

These consultations provided a valuable opportunity to observe how AI practitioners conceptualise system failures. A recurring theme emerged: participants, who often possessed strong technical backgrounds in fields such as computer science, mathematics, and engineering, demonstrated a sophisticated understanding of potential technological failure modes. This technical expertise aligns directly with the first category in our proposed taxonomy: 'Failure of the machine components'.

Conversely, the consultations revealed that the complexities arising within broader sociotechnical systems, particularly those with an (implicit) human-in-the-loop, were less consistently understood. The dialogues surrounding the limitations and potential failures of human–machine interactions highlighted critical areas for conceptual development, and showed that a more fine-grained taxonomy of failure modes is warranted than simply categorising all other failures as 'human error'. These interactions, therefore, were crucial in helping us develop essential insights, thus allowing us to identify the subsequent categories of our taxonomy.

The issues that arose with the startups consistently coalesced around four foundational themes beyond the purely technical. These themes, which informed the taxonomy's structure, included:

- Process and Workflow: For example, the design and strategic importance of robust data segregation practices, illustrated by questions concerning the rationale for utilising unique encryption keys for different clients to mitigate a (human) loss of a key.
- Human–Machine Interface: For example, the psychological and ergonomic significance of interface design choices, such as the colour and typography of warning messages, in conveying critical information.
- Human Component: For example, the necessity of anticipating and modelling a wide spectrum of user behaviours, including those that are unconventional or counter-normative such as the unanticipated uploading of sensitive information by users to an insufficiently-secured cloud storage system.
- Exogenous Factors: For example, the complex ethical and legal considerations involved in the contractual allocation of liability between software providers and B2B end-users.

A common theme in these 'mixed' failure modes was that startups, and in particular the technical developers themselves, did not necessarily see such failure modes as their areas of responsibility. Multiple times, the author heard variants of the statement 'This problem would be human error, and

not a problem in our technology, so we are not responsible for it'. The contrast between nuanced failure modes and the general idea of 'being a user problem', helped spawn the idea that with poor sociotechnical choices at the design stage, the humans participating in the HITL setups were being (unintentionally) *set up to fail*.

The fineness/coarseness of the taxonomy as 5 failure categories is a pragmatic interpretation of the topics that came up during these interactions, and how the various developers and managers were able to deal with them. We tried to keep in mind what appears to be best-suited to those designing, developing, and deploying HITL setups, and we note that there may well be other, more philosophically-oriented breakdowns one might use.

Many of the individual failure modes which were observed during these consultations can by now also be found in the wider literature, albeit there they are presented with a different structure and/or in unstructured ways. For example, the recent literature survey written by Sterz et al. (2024) develops a taxonomy for general human oversight in the context of AI. While they only consider the failure categories of technical design, human, and environment, they still discuss many of the individual failure modes presented in our taxonomy. We note that the differences between our taxonomy and that of Sterz et al. (2024) are a reflection of the empirical experience of conducting ethics consultations with startups, whereby discussions of the failure modes were naturally ordered by 'increasing human-ness' or 'lack of technicalities,' and many concerns emerged at the boundaries between human practices, abstract processes and technological details (thereby requiring us to have 5 categories). Moreover, Chiodo & Müller (2025) cover the harms and failure modes of general mathematical work in their '10 Pillars of responsible development', which also has close connections to our taxonomy and its breakdown into various failure modes.

Having studied failure modes for AI-supported governmental decisions, Green (2022) also notes that the introduction of a HITL setup can harm the overall safety of a sociotechnical system; something that was also observed during the ethics consultations, as the 'this is not a technical problem' attitude was often accompanied by placing a lot of potentially unwarranted hope on human oversight, thus creating a false sense of security in the AI product or service being produced.

Regarding the consultations with startups, the author also found that discussions on failure modes in automated industrial processes, as well as the commonly discussed 'Swiss cheese model' (Larouzee & Le Coze, 2020) for accidents, whereby an accident can happen despite many defensive layers if the 'holes' of each layer are properly aligned, were still relevant in HITL contexts. For example, Bainbridge (1983) wrote about the automation of industrial processes, Endsley (1995) studied the situational awareness of humans in automated processes, Sarter et al. (1997) looked at automation surprises, and Reason (1990) for human errors—the latter two have influenced categories 3 and 4 of our taxonomy, focusing on the human and the human–machine interface. Understanding that many of these failure modes are not unique to AI systems or more general ADMSs, but can also be found in other (industrial) situations, helped convey the spectrum of responsibilities to various startups. We present here in Table 1 an extended (but still non-exhaustive) breakdown of our taxonomy, with additional failure modes included, extending the list from §3.1. This is done in tabular form for ease of reading:

Table 1: HITL failure modes

| Failure of the machine components | Failure of the process and workflow | Failure at the human–machine interface | Failure of the human component | Exogenous circumstances |
|---|---|---|---|---|
| • Unexpected inputs or outputs
• Problematic machine evolution or self-adaptation
• Hallucinations
• Reasoning errors
• Overfitting of training data
• Biased or other erroneous outputs
• Unfalsifiable outputs
• Lacking 'common sense'
• Morally unacceptable outputs
• Other unexpected behaviour | • Insufficient power of the human
• Insufficient self-control/ independence
• Insufficient reaction time
• Unrealistic expectations
• Delayed notification
• Lack of disaster planning
• Insufficient management support
• Insufficient psychological support
• Lack of rest
• Conflict of interest
• Other process and workflow failures | • Incomprehensible or incomplete outputs
• Complex or poorly designed user interface
• Constantly changing user interface
• Insufficient training
• Poor documentation
• Transition failures between different humans
• Other HCI adaptability failures
• Other epistemic failures
• Other interaction failures | • Cognitive bias
• Automation bias
• Confirmation bias
• Fatigue
• Incongruous intentions
• Stress or overload
• Lacking courage
• Lacking motivation
• Lacking self-awareness
• Lacking humility
• Onset of groupthink
• Other human-centric failures | • Unreasonable laws
• Unreasonable societal expectations
• Conflicting requirements
• Misaligned objectives
• Political pressure
• Unexpected exogenous shocks
• Poor safety culture
• Inappropriate workplace requirements
• Insufficient resources
• Other external pressures |

One can further see very recent literature connected to each specific failure category given above:

1. Regarding failure of the machine components, Barassi (2024) investigates what the term 'AI errors' even means given the new and complex world of LLMs and other multi-modal models. And Kim et al. (2025) discuss how ADMS errors can systematically feed into human performance and human errors further down the line.

2. Regarding failure of the process and workflow, Rosenthal-von der Pütten & Sach (2024) conduct a deep investigation showing many humans in HITL setups may simply be unable to detect *systemic* bias in the overall output of an ADMS (in their study, a hiring algorithm).

3. Regarding failure at the human–machine interface, Tsai et al. (2025) have studied human performance on ADMS-assisted verification tasks, showing a substantial increase in cognitive processing and cognitive load on the humans when the ADMS assistance was activated due to the increase in content being displayed to them.

4. Regarding failure of the human component, Alon-Barkat & Busuioc (2023) demonstrate 'selective adherence' of humans, which is 'the propensity to adopt algorithmic advice selectively, when it matches pre-existing stereotypes about decision subjects' (Alon-Barkat & Busuioc, 2023, p. 154).

5. Regarding exogenous circumstances, Laux & Ruschemeier (2025) provide a critique of the EU AI Act, and in particular how its 'focus on providers does not adequately address design and context as causes of automation bias' (Laux & Ruschemeier, 2025, p. 1).

## C.2 Demonstrating a distinction between failure categories 2 and 4

There might be some ambiguity between failure category 2 (process and workflow) and failure category 4 (human component) of our taxonomy in §3.1, as each may seem to convey failure purely with the human in the HITL setup. But these are genuinely distinct, and there are examples where one of these categories failed but the other operated normally. We give some here.

Failure of the process and workflow (cat. 2), where the human component (cat. 4) did not fail:
As presented in Mattern (2007), during the Cold War the Soviet Union had several early-warning bunkers set up, scanning the skies for any incoming nuclear missiles in order to give central command time to launch retaliatory strikes. Lieutenant Colonel Stanislav Petrov worked at one such bunker, Serpukhov-15, near Moscow. On 26 September 1983, while acting as the duty officer there, the automated computer monitoring system there reported that U.S. missiles had been launched. This reported automatically to the early-warning system headquarters, which in turn eventually reported directly to Soviet leadership. Were a report that missiles had been launched to reach Soviet leadership, a retaliatory strike would have almost certainly followed, leading to a full-blown nuclear war between the two nations. Petrov made the decision to inform the early-warning system headquarters that the alert was a false alarm. He made this determination as he only saw five missiles on his computer screens and considered that it was unlikely a U.S. nuclear strike would be this small, and so must be an error (Hoffman, 1999). He was correct; the detection system was indeed erroneous. Had he not actively overruled the automated report in this way, a nuclear war may well have commenced. Clearly, the process and workflow of this HITL setup was problematic as messages were automatically sent to the early-warning system headquarters, and it was only on account of the extraordinary actions of the human component (Petrov) that a quick correction message was sent; Petrov had mere minutes to react, but did so effectively.

Failure of the human component (cat. 4), where the process and workflow (cat. 2) did not fail:
As presented in (DeMasters, 2023), on 29 April 2023 a tourist following GPS navigation drove their car into the ocean at Honokohau Harbor, Hawaii, and needed to be rescued (the car itself ended up completely submerged in the water). It was reported that 'the GPS they were following led them straight to the water'. The event occurred in broad daylight, and no contributing factors were reported apart from the GPS misdirection. The process and workflow of this HITL setup seems to have worked properly; the car was being driven by a human, and the GPS navigation system gave a *recommended* route. The human component (the driver) failed dramatically, as they did not use the context of their surrounds to scrutinise this recommendation sufficiently but instead deferred entirely to the automated recommendation, thus leading to the car being driven into the ocean.

## C.3 HITL as part of an AI security scanner setup

As presented in Robertson & Chwasta (2025), in April 2025 two men were alleged to have brought firearms through the ADMS-powered security scanning setup at the Melbourne Cricket Ground (MCG). This was an endpoint action HITL setup, with scanners running an ADMS used to flag attendees who *might* be carrying weapons, and then manual secondary screening used to conduct a final, definitive inspection. Initially, the blame was conjectured to have been 'human error' in the secondary scanning process. However, as mentioned in (ibid.), with a potentially high false-positive rate, secondary (human) scanners may have faced the combined challenges of operator fatigue (having to scan many attendees in a short space of time as they arrived for the beginning of a match), as well as complacency (having to scan countless attendees, *none* of whom were actually carrying weapons). The human scanners may have become quite tired, and many have lost faith in the ADMS (the scanner) as it overwhelmingly returned false positives. Indeed, shortly after initial reporting, it was revealed that the alleged offenders did not in fact have a metal detecting wand waved over them, even though the ADMS flagged them for additional checks (Ryan & Eddie, 2025). Both men were eventually prosecuted and found guilty of bringing firearms into the MCG (Cosoleto, 2025).

## C.4 HITL as part of semi-automated fire detection setups

As presented in Bennhold & Glanz (19.04.2019); Peltier et al. (18.06.2019), in April 2019 a catastrophic fire broke out in the attic space of Notre-Dame Cathedral. The cathedral had an ADMS for fire monitoring with an endpoint action HITL setup. When a fire was detected by the machine, a

short message 'fire' was sent to the monitoring office in the cathedral, without specifying exactly where the fire was (incomplete outputs). As is standard across France, fire alarms never automatically notify the fire brigade, so as to avoid false callouts (unreasonable laws). The employee on duty was only on his third day of the job and working unsupervised (insufficient training, unrealistic expectations). The employee was required to phone a guard, who then had to physically check the attic; a 6 minute journey up many stairs (insufficient reaction time). Unfortunately, the guard got lost, and went up the wrong staircase to the attic of the sacristy, which was the adjacent building (insufficient training). The employee then called his manager rather than the fire brigade (lacking courage), but could not reach him (insufficient support, insufficient power of the human), and it took time for the manager to call back and instruct the guard to leave the sacristy, go down the stairs, and then climb another staircase to the attic of the cathedral. By the time the guard reached the attic of the cathedral, the fire was raging. The fire brigade was then called, over 30 minutes after the first detection of a fire. This was a fairly simple system, computationally speaking, but nonetheless was a HITL setup that failed completely with spectacular consequences.

## C.5 MITIGATIONS FOR THE UBER HITL SETUP

In §3.3 we describe the trivial monitoring setup used by Uber in their self-driving car that was involved in a pedestrian fatality, where we signposted some of the failure modes from our taxonomy in §3.1 as they manifested in that case study. We now provide some mitigations that might have been taken (by Uber, or others) to address these and reduce the risk of a harmful outcome.

- To reduce fatigue, give more realistic human expectations, and involve the human more in the process, Uber might have created a squeeze trigger on the steering wheel, allowing the human to rest their hands on the wheel and 'feel' the AI driving, while not disengaging the AI unless they gave the trigger a gentle squeeze.

- To ensure vigilance and detect fatigue or distraction, Uber could have installed a vigilance device such as a light or buzzer that requires the human to press a button within a short period after notification. Uber could have also installed a dead man's switch, to trigger if the human took more than one hand off the wheel at once (working in tandem with the squeeze trigger suggestion above).

- To reduce fatigue, Uber could have set the human periodic tasks, such as to turn the windscreen wipers on/off, briefly change the intensity of the headlights, etc., rather than have them sit there with no set tasks for prolonged periods.

- To reduce automation bias, Uber could have implemented a system whereby the human and AI driving systems would swap periodically, say, every 5 minutes. That would have increased the sense in the human whereby they had definite responsibility, regularly.

- To compensate for machine failure in the form of misclassification of the pedestrian, Uber could have implemented some change point detection within the classification system; any rapid fluctuation between (re)-classification of an object within a short period (here, rapidly switching between 'another vehicle', 'pedestrian' and 'other' within 4.1 seconds) might imply some high level of uncertainty and, in such situations, trigger a warning to the driver to increase vigilance, and possibly even initiate a controlled slowdown. This could make use of an ML technique of *learning to defer*, where models are trained to defer to human judgement under certain circumstances. For supporting details of this method, see D.4.

- To reduce instances of the human choosing to carry out other tasks and thus be distracted, Uber could have carried out a security check of the human, to remove any distracting devices such as personal smartphones, and configured any Uber-issued devices to be unable to play videos or similarly distract the driver. Uber could have also provided the human with explicit guidelines, training, and examinations, to ensure that they were aware of, and competent in, avoiding distractions.

- To deal with the poor company safety culture, Uber could have changed its internal management and training systems, to teach, encourage, and reward, safety-conscious work. This could have included things such as safety training, regular safety checks of work (by management and/or peers), and a reward system (bonuses, certificates, etc.) for employees discovering unsafe aspects of the system.

- To encourage various actors to take on the responsibility of ensuring safety, the surrounding legal-regulatory system could have been set up in a way to make it clear who in the process would be accountable should an accident occur. A regulatory system where *some* accountability is placed on *all* parties might have (externally) encouraged more individuals, and Uber as an organisation, to take on more responsibility for ensuring safety and responding to concerns raised by whistleblowers. We explain one approach to a more distributed accountability model in §4.4 based on similar cases with complex causation chains where many actors contributed to the resulting harm that occurred.

Of course, this list is not exhaustive. And not every mitigation is guaranteed to work perfectly. However, by using our taxonomy from §3.1 to identify the potential points of failure of this HITL setup, *enough* of these failures could have been *sufficiently mitigated* to properly invoke the Swiss cheese model of accident causation and prevention (Larouzee & Le Coze, 2020), as already mentioned in C.1. A system does not need to work perfectly to prevent harm, but if too many subsystems operate substandardly, harm is inevitable. Our taxonomy from §3.1 gives a concrete tool to help avoid this happening.

## D HITL IN THE LAW

The main purpose of this appendix section is to provide additional details for the legal cases and arguments that we present in the manuscript, the details of which are far too long to fit in the main part of the manuscript. We now signpost this here for convenience.

We begin in D.1 with a brief explanation and justification for limiting our scope to EU and UK law. In D.2 we cover the legal concept of automated decision making, and how this relates to our definition of trivial monitoring. There, we detail the SCHUFA case, where the courts found that the the credit scoring company SCHUFA violated the GDPR by (inadvertently) carrying out automated decision making by adopting a trivial monitoring setup.

We then highlight the legal benefits of endpoint actions in D.3, showing that such HITL setups can help the human to carry out their legal and moral safeguarding duties to a better extent. There, we also show that SCHUFA could have averted violating the GDPR had it made use of an involved interaction setup. And in D.4 we go into detail about how existing machine learning strategies, such as learning to defer, can be integrated into our involved interaction HITL setups to improve operation and efficiency of the setup.

In D.5 we detail how causation can be difficult to determine in an involved interaction setup. Then, in D.6 we apply our legal and moral insights from our studied HITL setups to additional reduction types, showing what can be carried over, and giving a way to formalise these reduction types into new HITL setups.

And in D.7 we present the outcome of some legal cases related to mesothelioma where a principle of joint liability was used by the courts to bridge responsibility gaps, and how this might apply to involved interactions when trying to establish legal liability.

### D.1 OUR CHOICE OF LEGAL JURISDICTIONS TO COVER

This manuscript examines several legal frameworks that govern HITL setups. Some operate *ex ante*,[12] such as the EU/UK GDPR and EU AI Act, which apply regardless of whether a failure occurs because they regulate the 'effectiveness' and 'meaningfulness' of human oversight (see §4.1, 4.2). Others operate *ex post*[13] when a HITL setup fails and causes physical or psychological harm, and here we focus on the common law of negligence in the UK. We adopt a broad approach by analysing both ex ante and ex post mechanisms to provide a more comprehensive picture of the legal instruments at play in the development and deployment pipeline, while limiting our jurisdictional scope to the UK and EU for the three reasons outlined below.

The first reason is because of the EU AI Act and GDPR's global influence. The EU AI Act is the world's first comprehensive AI regulatory framework. Building on the influence of the GDPR, it is expected to have a 'Brussels effect', serving as a blueprint for other jurisdictions considering how to regulate AI (Siegmann & Anderljung, 2022). Gunst & De Ville (2021)'s work on how the GDPR 'conquered Silicon Valley' demonstrates how EU standards in data protection influenced state-level developments in the US, such as California's data protection law. New regulations introduced in the California Consumer Privacy Act have already mimicked the GDPR's Article 22(1) solely automated decision-making provisions by imposing obligations on businesses who use 'computation to replace or substantially replace human decision-making' (Ridgway et al., 2025). This is very similar to the ADMS definition in Article 22 of the GDPR and includes similar safeguards and obligations. It is also likely that provisions in the EU AI Act will translate to US state regulations as more companies internalise EU standards across their operations (Siegmann & Anderljung, 2022).

Siegmann & Anderljung (2022) argue that this dynamic is particularly relevant for large US-based technology companies whose AI systems fall into the Act's high-risk category (which is the focus of this manuscript). Although the EU's Code of Practice is only applicable to the provisions in the EU AI Act relating to general-purpose AI, major AI companies including Anthropic, Microsoft, OpenAI, Mistral AI, and Amazon have already signed the Code, signalling early intentions to comply with the Act's requirements. Therefore, in this manuscript, we focus on the EU jurisdiction because

---

[12]This means 'before the event', and refers to proactive laws that aim to prevent future legal violations.

[13]This means 'after the event', and refers to laws that punish actors after the legal violation has occurred.

of the GDPR and AI Act's broad applicability in all 27 EU member states, but in anticipation that its human oversight and automated decision-making requirements will have an extraterritorial scope.

Secondly, the authors acknowledge that the US and China are crucial jurisdictions to consider with respect to their regulation of HITL (in relation to the former, the Uber case §3.3 demonstrates this); however, analysis of these jurisdictions falls outside the scope of this manuscript. The US landscape comprises a complex patchwork of rules, executive orders, and legislation which all differ per state, making it difficult to analyse coherently within the constraints of this manuscript. China has taken a more targeted approach, with rules and standards based on specific products and services, for instance, TC260-003 on generative AI which incorporates HITL-related requirements for the labelling and detection of unlawful content (McWhirter & Tam, 2025). While both are important jurisdictions to consider, these regimes warrant a separate and more tailored examination, which would have diluted the in-depth legal analysis of the UK and EU frameworks if provided in this manuscript. Our focus on the EU AI Act and EU/UK GDPR should therefore be understood as a *proof of concept* for connecting HITL formulations, failure modes, and legal requirements that future research can extend to other jurisdictions. We invite further research, specifically as an obvious starting point on how California's new automated decision-making requirements will map onto the HITL failure taxonomy (§3.1) presented in this manuscript. As well as this, we suggest that future research also considers a comparative study of the legal oversight and accountability frameworks with respect to ADMS alongside the HITL taxonomy proposed in this manuscript (§3.1). To name a few jurisdictions that could be relevant, Brazil, Nigeria, and India.

Thirdly, this manuscript also draws on the UK common law of negligence to analyse how liability is assigned when HITL setups fail. Like the EU's legislative frameworks, UK common law remains influential beyond its borders. As former British colonies, many Commonwealth countries still retain foundational common law elements and principles from the UK in their legal systems today. For instance, in relation to causation in negligence (although not an exhaustive list of countries) India, Australia, Canada, and South Africa share the similar foundational doctrines and rely on UK case law as persuasive authority when making decisions. As such, the arguments advanced here for ensuring negligence principles account for HITL failure modes and avoid 'HITL scapegoating' by drawing upon mesothelioma cases (§4.4) will be relevant across these jurisdictions and contribute to broader conversations on common law approaches to AI accountability.

### D.2 TRIVIAL MONITORING AS AUTOMATED DECISIONS (GDPR)

At present, Article 22 of the GDPR generally prohibits trivial monitoring and classifies it as a solely automated decision despite the presence of an active HITL setup. The GDPR requires that any ADMS used to make decisions with legal (or similar) effects need to reflect at least an endpoint action HITL setup to 'break automation'. The scope of Article 22 was affirmed by the Court of Justice of the European Union (CJEU) in the SCHUFA case (Court of Justice of the European Union, 07.12.2023). Prior to SCHUFA, it was thought that if there was *any* active HITL setup then this could not be a solely automated decision within the scope of the Article 22 prohibition. However, in the case, the judge opined that a HITL setup *can* still exist within a solely automated decision if the human is just carrying out trivial monitoring. This is because in trivial monitoring, the human is not meaningfully influencing the decision-making process, and so is unlikely to have any impact on the ultimate output of the system, meaning that the decision-making is still automated.

The SCHUFA case concerned a credit scorer (SCHUFA) who created credit repayment probability scores through automated processing. SCHUFA then shared these scores with lender banks who determined whether they would lend money to individuals, using those scores. The issue in the case was whether an automated decision occurs between SCHUFA and the individuals, even though the bank lenders were in the middle of the decision making process (acting as 'the human' in this HITL setup). However, since the bank lenders relied so heavily on the scores, in reality, it meant that they were simply applying the output of SCHUFA's decision. Thus, the CJEU determined that this was an example of a solely automated decision between SCHUFA and the individuals seeking a loan, showing that trivial monitoring setups are generally considered to be automated if the human has no meaningful influence on the decision.

This case marked a significant shift in the law's approach to HITL by recognising that tokenistic humans cannot be used to enable companies to fall outside the scope of Article 22 and thus avoid

the onerous obligations associated with this provision. If data controllers want to perform an automated decision with no human intervention or if the human is just carrying out trivial monitoring, to be lawful, they must also have explicit consent from the decision subject (or another specific justification) that permits decision making with such limited oversight. Additionally, when performing automated decision making, the data controller must also have certain safeguards in place, such as the ability for individuals to request human intervention, receive explanations of decisions, and contest the outputs.

In order for a decision making process to fall outside the scope of Article 22 and 'break automation', the human needs to actively influence the decision making process. The CJEU provided little guidance on what the ideal HITL setup looks like since this was beyond the scope of the case. However, guidance from the UK's data protection authority (Information Commissioner's Office, 2025) suggests that to avoid making an automated decision, data controllers need to ensure that the human weighs up the output and interprets it before applying it to the decision subject. This reflects endpoint action, since the machine asks one real query and the human considers the machine's output and adds some more insight to make the final decision.

Therefore, the law recognises trivial monitoring as solely automated and generally prohibited (unless consent or other justification and safeguards are present). In addition, regulatory guidance suggests that endpoint action is required to move data controllers outside the scope of automated decision making in Article 22. However, as shown in this manuscript, there are significant yet distinct ethical and computational implications between trivial monitoring, endpoint action, and involved interaction setups that the law does not acknowledge.

We argue in this manuscript that involved interaction provides legal benefits that align with the broader responsibilities set out in the UK GDPR and EU AI Act. Therefore, building upon the jurisprudence of the SCHUFA case, data protection authorities should provide guidance to companies to implement at least endpoint action, and preferably involved interaction as an ideal HITL setup to fall outside the scope of Article 22. Existing guidance lacks practical tools that ADMS developers can use to enable *meaningful* and *effective* oversight. As such, including the specific computational setups like trivial monitoring, endpoint action and involved interaction with their applicable legal obligations would help clarify the scope of automated decisions to technical audiences and provide developers with concrete methods to configure their human–machine setups accordingly.

## D.3 THE LEGAL BENEFITS OF INVOLVED INTERACTION

Article 14 of the EU AI Act states that 'Human oversight shall aim to prevent or minimise the risks to health, safety or fundamental rights'. Similarly, Article 22(3) of the GDPR stipulates that data controllers implement 'suitable measures to safeguard the data subject's rights and freedoms and legitimate interests'. These safeguarding duties are onerous and it is difficult to understand how the human in a trivial monitoring or endpoint action setup would be able to minimise risks to an individual's human rights or safeguard a decision subject's legitimate interests. In trivial monitoring and endpoint action setups, the human may have no influence on, or understanding of, the computational process involved in the decision.

Indeed, in these setups the human may face a completely 'black box output'. For example, in an endpoint action, the human may just receive a probability score for credit risk. Using only this, the human would need to determine whether it should be applied to an individual and also whether it infringes any values. Thus, endpoint action and trivial monitoring setups contain an inherent opaqueness which prevents the human from understanding the significance of certain inputs or factors on the output of the ADMS. As a result, the human may lack sufficient knowledge and ability to influence, effectively evaluate, and implement the decisions made by an ADMS to uphold their safeguarding duties.

In §4.2, we argued that in systems with weaker reductions such as endpoint action, the human is not enabled to effectively influence the system, and thereby meet their obligations under GDPR or the EU AI Act. We therefore propose that involved interactions are included in regulatory guidance to guide developers on how the human can meaningfully impact the system, thereby meeting legal requirements.

Carrying forward the example of the SCHUFA case (D.2), if the lender was engaged in an involved interaction setup, the ADMS could query the human in the course of the computation which would enhance the human's understanding of whether the decision subject's rights or legitimate interests are being eroded. For instance, the human could be asked questions by the ADMS such as 'is the individual's postcode relevant to the credit score?' or 'what information should be used to substitute the lack of credit history?'.

Additionally, involved interaction setups give the human a chance to watch and scrutinise more of the computational process, as mentioned earlier. This may help overall safety, as they may 'spot a problem' early on, such as the ADMS asking questions about a credit applicant that should not be used as part of the assessment. Therefore, involved interaction avoids the 'sloppy' (Crootof et al., 2023, p. 434) implementation of a human into human–machine setups, by enabling the human to better fulfil their safeguarding duties by evaluating the output of an ADMS in accordance with the rights and freedoms of decision subjects.

Also, an involved interaction setup could have actually *averted* SCHUFA's legal issues regarding the court's determination of their system as automated decision making (prohibited by Article 22). In that case, it was deemed that even though the intermediary bank *appeared* to be acting as 'the human' in an endpoint action HITL setup by taking the credit score from SCHUFA and using it within their own processes for determining credit worthiness, the courts determined that in actual fact that the human was in a trivial monitoring setup as they were merely 'passing on' the credit score as a credit worthiness determination. What we see here is that in this ostensibly endpoint action setup, the human 'slipped back' to a trivial monitoring setup by completely deferring to the machine output; a clear case of (a particular form of) automation bias in our taxonomy of HITL failure modes (see §3.1); one which was deliberate and designed into the system which unintentionally resulted in serious legal implications. Had the human (the bank) properly maintained its (endpoint action) role of using the machine output to feed into a human decision, then Article 22 of the GDPR would not have been violated by SCHUFA.

However, as explained in §3, HITL setups naturally fail, for a wide variety of reasons (automation bias being one of them), and this should have been anticipated by SCHUFA. So what could SCHUFA have done to avert violating Article 22? One way would have been to properly implement an involved interaction setup, needing several interventions from the human in the machine computation. This would have ensured the decisions were not automated, thus avoiding a violation of Article 22. With the endpoint action setup SCHUFA had in place, the human could simply 'nod through' the machine output in an algorithmic way, by accepting all applications scored above a certain threshold, and rejecting all those below it, meaning the setup was given by a total function. As such, this broke the requirement of an endpoint action, which is that 'the oracle machine many-one reduces the computation to the human, but it *does not* define a total function' (§2.1).

If, on the other hand, the setup was as involved interaction, the human could not have nodded through the machine queries throughout if they were indeed real queries (as defined in §2), as they may well have faced a query not admitting a yes/no answer. Of course, the queries might all be algorithmically solvable (and so the setup again reduces to an involved interaction), but here SCHUFA would have been *incentivised* to ensure this did not happen, to avoid violating Article 22. Thus, SCHUFA would have been in a much better position to avoid carrying out automated decisions (i.e., avoid violating Article 22) had they implemented an involved interaction setup whereby the human (the bank) needed to answer questions throughout the computation process. By implementing an involved interaction (§2.1) to break the automation bias of the human (§3.1), SCHUFA could have prevented the slip back to trivial monitoring and thus avoided violating Article 22 (§4.1).

This demonstrates a real example where our arguments from §4.2 play out, illustrating the benefits of having a HITL setup with stronger reductions, and thus more human involvement and input.

## D.4 HITL AND ITS CONNECTION TO DIFFERENT LEARNING STRATEGIES

The concept of *learning to defer* (L2D), as introduced by Madras et al. (2018), describes an adaptive learning framework within a two-stage sequential socio-technical system. Here, an AI model is trained to either generate a prediction autonomously or to defer the decision to a human decision-maker. This policy is adaptive: the model considers not only its own confidence but also the anticipated performance of the human decision-maker. This is in contrast to the simpler concept of

*rejection learning*, where the model defers to a human when it is not confident in its own prediction, independently of how competent the human may (or may not) be. In L2D, if the model chooses to predict rather than defer, the human is completely bypassed, and the model's output is adopted as the system's final decision.

According to the formalisation presented in §2.1, the standard L2D setup does not constitute a HITL setup. Our formalisation conceptualises HITL as a socio-technical system where the human has guaranteed influence or oversight capability (from the ability to stop the machine, to being properly involved in the decision-making process). However, as described by Madras et al. (2018), L2D violates this requirement because the machine has the authority to exclude the human from the loop entirely. Additionally, in L2D systems the human's agency is eroded by the system considering a computationally produced analysis of both the human and itself. Similarly, rejection learning either eliminates or engages the agency of the human, but only by considering a computationally produced analysis of itself. In both cases, in general the human's agency has been effectively eliminated. While there will be cases for the human to potentially affect the output of the whole system, agency was defined as the human's '*ability* to actively impact the system' (§2 and B.3), and in general the ability of the human to impact the system is contingent on the machine's computational analysis. Because of this contingency, these setups do not place the human in a position of agency because their ability is always contingent on factors that they have no control over. Computationally, L2D implements an oracle machine that executes *at most one* real query. If the machine chooses not to defer, it executes zero queries, operating as a human-out-of-the-loop (i.e., HOOTL) setup.

Despite not fitting the strict formalisation of a HITL setup, L2D offers significant utility in many contexts. As Madras et al. (2018) argue, an autonomous decision made by a machine may be preferable in certain circumstances. For example, when it demonstrably outperforms the human (e.g., specific medical diagnostic tasks), when the human decision-maker exhibits systematic biases that the model can mitigate, or in high-volume or fast-paced environments where human (cognitive) capacity or stress resistance are limiting factors. Thus, a genuine L2D setup can help mitigate certain failures of the human component (§3.1). Alternatively, providing the machine with the ability to defer a task also helps mitigate failures of the machine component (§3.1). One specific instance where L2D could have improved a HITL setup is given in the discussion of potential mitigations for the Uber case in C.5,

However, giving the machine the authority to exclude the human entirely can violate legal and regulatory requirements for specific HITL setups, as well as open up issues of moral responsibility given the erosion of human agency. If the machine were to carry out an entire decision process with no human intervention, then this would fail to satisfy requirements in the EU AI Act which require effective human oversight for high-risk AI systems, as described in §4.1. In addition, this would be classified as a prohibited solely automated decision as per Article 22(1) of the EU/UK GDPR, requiring an exception like explicit consent to be lawful. Even if the machine had the *option* to defer to the human, if a particular decision was made *without* such a deferral then that would still remain a 'solely automated decision'. Furthermore, as discussed in the SCHUFA case (for supporting details see D.2), merely adding a human to approve the machine's decision to not defer wouldn't alleviate this. The argument that the machine decided not to defer because it believed it had a better chance of making a good decision than the human would not be a justification for making such an automated decision, even if it could be shown statistically that the machine was more 'reliable' on average; the GDPR is concerned with *individual rights* relating to the processing of personal data, not *average-case outcomes* of the decisions made in automated processing.

The effective elimination of the agency of the human, while sometimes enabling the human to impact the system—that is, eliminating the human's *general* ability to impact the system, while *sometimes* enabling it to do so depending entirely on the machine's analysis—presents a significant moral issue. One of the benefits of an involved interaction setup is that it enables the human to align their values with the machine. However, in this context the human's influence being contingent on the machine's determination implies that any values the machine may 'embody' by virtue of, say, its training data, will supersede any values the human may input. This defeats the objective of value alignment, which would require the values of the human to supersede that of the machine. Further, such a setup also substantially enhances the practical risk of scapegoating, where systems appear to have a human in the loop, but in reality these humans have no real agency over the system whatsoever, but yet are held responsible for any harmful outcomes. Similar systems that allow the machine to disregard the input of the human on any computationally designed basis run into the same fundamental issue with

value alignment, where the values of the machine will supersede that of the human. Any system that *eliminates* the agency of the human, restricting their impact on a computationally designed basis, is therefore not seen as a positive setup either morally or legally.

In such situations, L2D is not a stand-alone HITL setup according to the laws mentioned above, or to our formalisation, but rather needs to be *integrated* into HITL setups, slightly differently from what Madras et al. (2018) imagine. Rather than giving the machine the option to exclude the human, the human needs to have the option to stop the computation (trivial monitoring), or the machine's output must pass through to a final human decision (endpoint action), or the human must properly interact with the machine to jointly improve the answer (involved interaction). Another example, where such adaptations would need to be made, is in the automated cancer diagnosis process discussed by Madras et al. (2018, p. 1), as under the EU AI Act's human oversight provisions in Article 14, the doctor would not be allowed to automatically accept the machines' diagnosis without effective oversight to check its validity.

But these legal restraints do not diminish the value of L2D; quite the opposite. L2D can become a crucial component in highly complex HITL setups such as involved interactions where the machine needs to determine various points in the computational process to ask for (i.e., defer to) human judgement or input. In the 'ping-pong' process between human and machine in an involved inter-action, L2D becomes an excellent method for the machine to understand when to pause and seek useful human input. If embedded within an involved interaction HITL setup, L2D presents a path towards operationalising this setup type which is otherwise difficult to achieve. This also goes some way to ensuring that the machine does not simply ignore human inputs; it is explicitly trained to de-fer to, and make use of, human inputs in certain circumstances. Embedding L2D, or even rejection learning, in an involved interaction setup can help make it practical at scale by giving the human a manageable load of queries to answer or inputs to make. This will reduce certain failure modes from §3.1 such as human fatigue, automation bias, stress, and insufficient reaction time. This would still adhere to legal requirements for human oversight and automated decision-making because the human still remains effectively *in the loop*.

More generally, as Ruggieri & Pugnana (2025, p. 28684) note, '[i]n many high-stake domains, abstaining from providing an output is a better strategy than bearing the risk of wrong outputs.' The ability to deal with aleatoric uncertainty (irreducible stochastic variability for outputs given the same input) and the epistemic uncertainty (e.g., data points which are substantially different from those seen during training) are crucial for any HITL setup. The formalisation presented in this manuscript does not assume that the human response is necessarily correct or better than the machine (for supporting details see B.1), and, as discussed by Madras et al. (2018), that does not always have to be the case. Hence, machine learning approaches which factor in limitations in the knowledge of the human and/or machine are an important aspect to consider when trying to operationalise the formalisation presented in this manuscript.

Ruggieri & Pugnana (2025) survey different such approaches, including rejection learning, dynamic model selection, learning to defer, and uncertainty estimation. However, as discussed in the context of L2D, not all methods presented in the literature represent genuine HITL setups according to our formalisation: often the socio-technical setup will need to be adjusted to *actively* include the human, or to at least give the human the option to intervene and interact with the machine. All methods surveyed by Ruggieri & Pugnana (2025) have the ability to provide valuable new information to the human. Such information ranges from knowing how close a data point is to the decision boundary (in ambiguity rejection), to the newness of a data point (in novelty rejection), to estimates about the relative human-machine strengths (in L2D), or more general information about uncertainties. Criti-cally, however, a HITL setup does not see this output as the result of a one-step machine procedure, rather it is seen as information on which *the human* should be able to act.

Such a perspective, however, does not explain, on its own, how the human can build trust in the machine and how the degree of trust should impact action. To solve this problem in the context of multi-label classification tasks, Straitouri et al. (2023) develop a HITL setup to improve expert predictions using *conformal predictions*: instead of presenting the human with an assortment of information about a data point and its classification, the machine creates a set of potential labels from which the human has to choose. Their implementation represents an endpoint action: the machine 'forcefully asks experts to predict labels from these sets' (Straitouri et al., 2023, p. 1). The machine reduces and thereby defines the set of available options, but the human has to make the final

decision by picking a label from the machine-generated set. Unlike in the L2D scenario presented by Madras et al. (2018), the human involvement is not optional in this setting.

Finally, the practical implementation of these learning strategies requires a careful balancing act between the different failure modes. As discussed, L2D has the ability to mitigate certain failures of the machine and/or human, but it can introduce new failures at the process and workflow level by giving the machine the ability to exclude the human from the loop. And conformal predictions may introduce additional automation bias, if the human is not allowed to consider or stops considering other labels or responses outside those suggested. Generally, the approaches surveyed by Ruggieri & Pugnana (2025) are designed to mitigate specific failure modes, but their successful implementation in HITL setups may require trade-offs to adequately deal with all five failure categories presented in §3.1.

## D.5   WHY LEGAL CAUSATION IS DIFFICULT IN INVOLVED INTERACTION

Involved interactions have a somewhat unfortunate drawback, in that they can make determining causation difficult. On the one hand, the reduction methodology we define in §2.1 does give a consistent and formal mode of analysing HITL setups in order to assess the extent of the human's involvement in the computation, and the meaningfulness of their involvement. Particularly, where the human can influence computational systems through real queries, it can be said that they have more meaningful involvement. The consistency of this approach, and the level of formalism presented in §2.1 (and justified in §2.3 and B.9), is also crucial for regulators in practice, enabling them to identify and address tokenistic HITL setups where the HITL has no meaningful involvement even though it may 'appear' as though they are doing a lot.

On the other hand, in an involved interaction, it becomes extremely complex to determine whether the human actually influenced the machine's computation throughout the 'ping-pong' process. Even if interaction logs are kept to trace the queries and back-and-forth interactions, it may not be possible to evaluate the impact of the human's input on the output of the ADMS due to the indeterminism of the computation tree. Each human input changes what the ADMS does next, and each ADMS query changes how the human views the process and thus how they might answer. It may become impossible to discern how and where failures arise and what these features are attributable to. Nevertheless, even if causation can be difficult to establish, in the law, this is not new. In D.7, we show how the courts have previously dealt with cases where the cause of harm is difficult to discern. As such, we suggest that these cases should be drawn upon when exploring how to determine liability in cases involving HITL failures.

## D.6   USING AND FORMALISING INTERMEDIATE HITL SETUPS

In B.4 we mentioned three intermediate reduction types, from which one could potentially also form HITL setup types. These are reductions that sit 'between' many-one reductions and Turing reductions, where each type has more reliance on the oracle (i.e., 'asks more queries', loosely speaking) than the previous. This presents a 'sliding scale' of five reduction types that one could consider,[14] with progressively more oracle involvement along the scale. Our discussion in B.4 gave some technical reasons as to why we did not carry out a full analysis of these intermediate reduction types in this manuscript, focusing on the fact that many-one and Turing reductions were at the two ends of that scale. We now explain why, from a legal and moral perspective, our existing analysis of how many-one and Turing reductions are used in HITL setups extends to provide insight into these intermediate reductions.

To begin, our legal analysis of the concepts of meaningful and effective oversight as they relate to the EU/UK GDPR and EU AI Act (§4.1) showed that, as far as these two regulations are concerned, endpoint action and involved interaction HITL setups[15] are *equivalent* under those laws. That is, a (very well implemented) endpoint action HITL setup might already satisfy the (current) legal requirements of meaningful and effective oversight (cf. Sarra (2024), as explained in §4.1). We argued

---

[14]These are: many-one reductions, bounded truth-table reductions, truth-table reductions, weak truth-table / bounded Turing reductions, and Turing reductions, where each reduction type implies all the types after it. So according to our interpretation of the *strength* of a HITL setup (B.6), this ordering is reversed when considering the strength of each of these as a HITL setup.

[15]Corresponding to many-one reductions and Turing reductions respectively.

that an involved interaction setup might *better* satisfy such legal (and moral) oversight requirements in certain circumstances (§4.2), and moreover that having an involved interaction setup might reduce the chances of accidentally 'slipping back' to a trivial monitoring setup (§4.2 and D.3). But, in terms of oversight obligations, the EU/UK GDPR and EU AI Act do not draw any explicit distinction between endpoint actions and involved interactions; these reductions (and thus all intermediate reductions between them) collapse down to the same legal category. Hence, from a legal perspective *on oversight* in ADMSs, considering these intermediate reduction types does not add much insight (with regards to compliance with the laws we are considering).

However, these same laws (the GDPR and EU AI Act) impose various safeguarding duties on the human, which (in §4.2) we argued can be partially realised through a HITL setup where the ADMS asks the human more frequent queries, thus giving the human more insight into the computational process, and so improving overall *explainability*. And while these laws do not specify the need for involved interactions directly, moving towards involved interaction setups can, as we showed in §4.2, improve explainability by virtue of the fact that they have more human queries than, say, an endpoint action setup. This, however, comes with an unavoidable trade-off as we discuss in §4.3: increasing the level of human involvement with clearer intermediate computational steps necessarily decreases the ability to assign responsibility for harmful impacts. Further inspection reveals that this argument is not specific to involved interactions; *any* increase in real human queries will increase explainability, but at the same time decrease the ability to assign responsibility. Thus, by examining the 'two ends of the spectrum' (endpoint actions and involved interactions), we have examined the two extremes of this trade-off. Insofar as the intermediate reductions mentioned above are concerned, ADMS developers, or indeed regulators, can now pick, or 'tune', this trade-off to suit the surrounding circumstances, and go up (or down) this 'sliding scale' to find the right *balance* between explainability and assignment of responsibility. For example, in certain sectors, establishing responsibility might not be a significant issue because insurance obligations will take effect and consume costs from a practical point of view (e.g., in medical practice, and in particular hospitals, it might be preferable to have more explainability over clearer accountability). And in other, more time-restricted situations, one might want less 'ping-pong' between the machine and the human to save time, ideally with very well-chosen points in the computation where the machine requests human input (as discussed in D.4).

By explaining *how* this trade-off operates, exemplified through our analysis of the two extremal reductions (many-one and Turing), we thus empower both developers and lawmakers to *choose their own* intermediate reduction, and form a HITL setup from it, on a case-by-case basis in accordance with the one that best reflects the needs of their system and the surrounding circumstances. Our explainability-responsibility analysis holds for all these intermediate reductions, and so it is merely a case of 'going along the scale' and choosing a reduction that is suitable. In essence, our analysis covers all possible reduction types between many-one and Turing. And this need not be restricted to the three intermediate types already mentioned; it is more general than this. Take any (possibly infinite) totally ordered set of (oracle machine) reduction types between many-one and Turing; our analysis will always apply to the HITL setups produced from such a set. Indeed, for any pair of (oracle machine) reduction types $P, Q$, if every reduction of type $Q$ is also of type $P$, then a HITL setup based on $P$-reductions that are not $Q$-reductions will always be further along the explainability-responsibility trade-off scale than one based on $Q$-reductions (in the direction of more explainability / less responsibility). Thus, if one can find or conceptualise a new reduction type that sits between any of the five we have listed so far (and locate where), then one can quickly see where a HITL setup built from it sits on the explainability-responsibility trade-off scale. Hence, from a legal perspective *on explainability*, dealing with these intermediate reduction types is already fully covered by our existing analysis.

There is a technical subtlety here that would still need to be overcome, which we now explain. When we formalised endpoint actions and involved interactions from many-one reductions and Turing reductions, we placed additional lower bounds on the minimum number of queries the machine could ask, and moreover forced these to be real queries. This was for several reasons:

1. To prevent a HITL setup based on a many-one reduction being considered as an involved interaction, as mathematically a many-one reduction is also a ('lazy') Turing reduction that only asks one question of the oracle.

2. To compensate for the large 'gap' between many-one and Turing reductions. The definition 'stronger than a many-one reduction' is too broad, as a reduction that always asks two queries in succession is stronger than many-one, but nowhere near as strong as a general Turing reduction.

3. To disregard queries that did not affect the computation, so that we had a better grasp of how many good (i.e., real) queries were being asked.

To carry out this formalisation for the three additional intermediate reductions types mentioned above would be a significant undertaking, and moreover would not address any additional reduction types that might be introduced later. So instead, we give a general method for carrying out a formalisation of reduction types as HITL setups as follows:

Take any (possibly infinite) totally ordered set $S$ of (oracle machine) reduction types between many-one and Turing, ordered by implication with ordering denoted $<$. With the notation of $P$-reductions given above, we say that a HITL setup is a $P$-setup (relative to $S$) if both of the following hold:

1. The setup can be described by an oracle machine carrying out a $P$-reduction to the oracle.

2. For every $Q \in S$ with $Q \lneq P$, the HITL setup cannot be described by a $Q$-reduction.

So a HITL setup is a $P$-setup if it can be simulated by a $P$-reduction, but not by any reduction $Q \in S$ weaker than $P$. One can then apply our analysis from above to this set $S$ of reduction types (and corresponding HITL setups). This still forms a sliding scale of increased reliance on the oracle (the human), and also reflects the explainability-responsibility trade-off. Note that, once this set $S$ is chosen, the concept of a $P$-setup is then well defined *relative to $S$*, but does depend on $S$: given two sets $S, S'$ that contain a fixed reduction type $P$, if $S'$ contains a reduction type $P'$ such that $P' \lneq P$ and $Q \lneq P'$ for all $Q \in S$, then the notion of $P$-setup will be stronger relative to $S'$ than relative to $S$.

With a set $S$ of reductions that is sufficiently large (five, in what we have described above), one can then argue that there is sufficient granularity to ensure that sliding along the scale leads to a genuine and meaningful increase in human involvement in the setup. This does not apply when $S$ consists only of many-one reductions and Turing reductions, as the gap is too large, which is why our earlier analysis uses a slightly different notion. And so, with the above formalisation, one could take any (possibly infinite) totally ordered set $S$ of reduction types and assign those as 'the reductions to choose from' (and then form HILT setups from).

A worst-case scenario here is that a $P$-setup ends up being 'a tiny bit better than a $Q$-setup', where $Q$ is the strongest reduction type below $P$. But with enough reduction types in $S$, this becomes less and less significant, as the 'gaps' between reduction types become smaller and smaller. It is mitigated entirely if no such 'strongest' $Q$ below $P$ exists, meaning that there are infinitely many reductions types in $S$ and they are *dense*[16] (with no discrete/isolated reduction types within it). At the implementation level, the difference between individual $P$-setups (i.e., 'gaps') stemming from $S$ would become negligible when $S$ is large enough; if $S$ were infinite and dense then no such gap would exist. However, with $S$ infinite and dense, verifying one had such a $P$-setup could then become quite hard, as there would be no 'next weakest' reduction type to compare to, and so one would probably need to approximate it by reductions 'sufficiently close' to $P$; such an approximation should suffice for practical purposes.

Though this generalised formalisation method does not exactly match how we formalised HITL setups from many-one and Turing reductions in §2.1, it is nonetheless a valid and motivated way to make use of large set of reduction types when considering HITL setups. Laws, technology, and societal desires may evolve over time, and our methodology here retains at least some persistence if further, more nuanced reduction types and corresponding HITL setups are deemed necessary in the future.

---

[16]The authors are unaware of whether an infinite set of totally ordered reduction types (ordered by implication) between many-one and Turing exists, let alone one that is dense and/or contains the three intermediate degrees given above. While not necessary for the further development of this manuscript (as there might be minimal practical value in having infinitely many reduction types to choose from), it nonetheless poses an interesting theoretical problem in mathematics for further work.

### D.7 MESOTHELIOMA CASES AS A SOLUTION FOR INVOLVED INTERACTION

In this manuscript, we raise a problem concerning the assignment of responsibility in an involved interaction setup. Here, we explore how the law can confront this challenge by looking back to previous cases which have dealt with similar 'responsibility gaps' in different contexts. The UK courts have previously departed from established causality principles to compensate workers who had developed deadly mesothelioma from exposure to asbestos fibres across multiple employers (House of Lords, 2006). In this line of cases, claimants had worked for several employers. At any point in their employment, they could have been exposed to asbestos fibres, causing them to develop mesothelioma. However, even a single asbestos fibre can trigger mesothelioma. As such, it was medically impossible to pinpoint which employer exposed the claimant to the asbestos that caused the mesothelioma. Departing from common causation principles in the mesothelioma cases, the court held that all the employers could be *jointly liable* if it could be proven that they 'materially increased the risk of harm', rather than finding a direct causal link between the employers and the mesothelioma.

In order to distribute the liability jointly, each employer's contribution to the claimant's overall risk of contracting the disease was considered. Courts sometimes divided the liability in proportion to the duration of exposure, even though they might not have actually caused harm directly. For example, if a claimant worked at one employer for 30% of their 'total exposure time', that employer was liable for 30% of the damages. This approach was confirmed in cases like House of Lords (2006) and later modified by the UK's Compensation Act 2006, which allowed claimants to recover full compensation from a single employer, who could then seek contribution from others.

The reasoning in the mesothelioma cases has received significant criticism from some legal academics since these cases were fundamentally decided on public policy grounds, as opposed to legal principle (Morgan, 2003). The judges in the House of Lords reasoned that it would be unfair not to compensate workers just because causation could not be attributed according to the existing legal tests. However, as shown in this manuscript, there are similar compelling public policy grounds to avoid the 'scapegoating' of the human where responsibility gaps emerge in HITL cases, such where involved interaction setups are employed. For example, although decided in accordance with US law, the Uber case described in §3.3 shows how the human can face injustice if the liability is solely attributed to them. The Uber case involved numerous failure modes which spanned across the taxonomy presented in this manuscript and beyond the control of the human (see §3.3). As such, when HITL setups fail, liability must be more distributed to account for responsibility gaps.

The mesothelioma cases are a good foundation for judges deciding cases involving the entangled human–machine configurations in involved interactions because they reflect the impossibility of determining causation. In principle, the judges could hold any of the contributors that materially increase the risk of harm in the HITL setup liable. In an involved interaction setup, it may be impossible to determine what actually might have influenced the harm, but a myriad of factors could materially increase the risk of harm. Such factors could include any feature of the system that significantly enhance the likelihood of HITL failure modes in §3.1, such as lack of training of the human, overworking the human causing fatigue, or even extreme bias in the training data used for an ADMS. Drawing upon the law's approach to liability in mesothelioma cases, in the context of this manuscript, identifying contributors which 'materially increase the risk of harm' can be used by the courts to hold multiple actors (including the technology companies) accountable so that the human is not treated as a 'scapegoat' solution in 'moral crumple zones' (Elish, 2019). Such legal solutions also incentivise companies to actively design systems to reduce the risk of failure because of the high liability consequences.

