# OpenReview forum: "Formalising Human-in-the-Loop: Computational Reductions, Failure Modes, and Legal-Moral Responsibility"
_ICLR.cc/2026/Conference — ICLR 2026 Poster_

### Official Review · Reviewer_LN6b · 2025-10-21

**Soundness:** 3
**Presentation:** 3
**Contribution:** 3
**Rating:** 6
**Confidence:** 3

**Summary:**

The paper formalizes human-in-the-loop (HITL) systems by mapping three configurations to concepts from computability theory: trivial monitoring (the human can only stop the process; the system is a total function), endpoint action (a single real human query; a many-one reduction), and involved interaction (potentially unbounded real human queries; a Turing reduction). It unifies related terms such as HIC (human-in-command)—a governance stance where humans retain ultimate authority over system goals and deployment—and MHC (meaningful human control)—designing systems so humans have timely, informed, and effective influence over critical outcomes—into one coherent framework, showing how this helps regulators identify tokenistic oversight and connect failure modes to specific designs. The legal discussion, focusing on EU and UK law, links these configurations to the GDPR and the EU AI Act, warning against “moral crumple zones,” where humans are held responsible for system failures beyond their control. Overall, the paper argues that meaningful oversight requires moving beyond trivial monitoring to designs that incorporate genuine, well-timed human queries, balancing explainability with responsibility.

**Strengths:**

Interesting case studies that trace multi-category failure cascades—bridges theory to incidents.

Novel and Rigorous Formalization: The application of oracle machines and computational reductions to formalize HITL systems is genuinely creative and mathematically sound. This provides a precise vocabulary for what has been a vague concept, distinguishing trivial monitoring, endpoint action, and involved interaction in a principled way.

Regulatory relevance: concrete reading of GDPR/EU AI Act and the SCHUFA precedent; shows how designs can slip back into “solely automated.”

Honest trade-off analysis (explainability vs. responsibility) with policy-minded remedies to avoid human scapegoating

Overall it was an engaging read even if not a standard paper and worth thinking more about.

**Weaknesses:**

Compelling case studies that trace multi-category failure cascades—bridges theory to incidents.

Legal scope: law discussion centres on EU/UK; portability to other regimes is not fully analysed.

Hard to translate this into real life: proving a system truly asks real queries (vs. veneer) is hard; guidance for auditors could be deeper.

Human-as-Oracle Limitations: While B.1 addresses this, the fundamental modeling of humans as fixed oracle functions is problematic. Humans learn, adapt, get fatigued, have strategic behavior, and exercise agency in ways that don't map cleanly to mathematical oracles. The formalization may obscure important sociotechnical dynamics.

Limited Empirical Validation: The taxonomy is based on consultations from 2020-2022 with primarily startups. There's no systematic evaluation of whether: (a) developers find this framework useful, (b) regulators would adopt it, (c) it actually prevents failures when applied. The framework is proposed but not validated as effective.

Limited validation that the framework actually helps prevent failures or improve systems

For an ML conference, limited technical ML content or empirical experimentation
No learned models, training procedures, or optimization
No experiments, datasets, or empirical ML evaluation
"AI systems" discussed are abstract ADMSs, not modern ML systems (in fact ended up googling if Notre Dame was AI based system)

**Questions:**

How can regulators or auditors verify whether a system truly involves “real human queries” versus superficial or scripted checks?

How might involved interaction be made practical at scale, especially in high-volume or time-sensitive contexts (e.g., content moderation, autonomous driving)?

Do you envision technical audit tools that could automatically classify systems by your HITL taxonomy (e.g., detecting trivial monitoring vs. involved interaction)?

How does your framework prevent systems from adding meaningless queries to appear as involved interactions while remaining functionally automated (like the SCHUFA concern)?

How should designers determine the minimum necessary level of human involvement that remains legally and ethically “meaningful” without creating inefficiency?

---

> ### Author Response · Authors · 2025-11-21
> **Review response 1/4**
>
> We thank the reviewer for their feedback, and we strongly agree that such theoretical work must be complemented by a more practical perspective. We have attempted to address these concerns by clarifying existing formulations, as well as extending and adding new appendices.
>
> # Responses to weaknesses.
>
> ## W1 Focus on EU/UK law.
> We thank the reviewer for asking for further justification for our jurisdictional focus. We agree that this should be fully explained in the manuscript. Our manuscript shows, as proof of concept, how mathematical formalism, failure modes, and law can come together to create a coherent system to create and analyse HITL setups. The manuscript focuses on EU and UK laws for three reasons:
>
> 1) The “Brussels Effect”: the EU AI Act is the world’s first comprehensive approach to regulating AI. We argue that, similar to the EU GDPR before, it can be reasonably expected that it will serve as an inspiration for other jurisdictions.
>
> 2)  The UK’s common law of negligence acts as persuasive authority in Commonwealth jurisdictions, such as Australia, Canada, India, and South Africa. Our analysis of case law on causation and HITL failures is therefore not only limited to the UK, but is applicable more widely across the globe.
>
> 3) While the US/China are forerunners in AI development and therefore crucial from a legal perspective, they present additional challenges which warrant a separate dedicated legal analysis: the US landscape comprises a patchwork of rules through a complex web of state and federal law, and China often uses a targeted approach for specific products and services.
>
> We added a full justification of our scope of analysis in Appendix D.1.
>
> ## W2 Guidance for auditors.
> We agree that further guidance for auditors is needed, as well as for developers. We have expanded Appendix B.9 to address this. There, we provide measures that developers can carry out to make their ADMS HITL setup more amenable to scrutiny and checks, as well as what auditors and other “checkers” can do to carry out such checks. We propose a framework whereby developers and checkers each shoulder some of the burden of ensuring the HITL setup can, and will, be properly checked.
>
> ## W3 Limitations of the concept of human-as-oracle.
> While humans are unique and their essence and behaviour cannot be fully captured through mathematics, oracles still provide a logically sound and useful framework for analysing their role in HITL setups. We have subsequently clarified our definition of the HITL setup types in Section 2.1. We agree that our formalisation needs to be applied with care as it contains a number of subtleties. As we discuss in the third paragraph of Appendix B.1, the human can change their opinions and even their values, become fatigued, or even physically interchange (as in, a second human can take on their role in the system). Further discussion of how humans can “fail” is given in Section 3.1, with additional details in Appendix C.1. The dynamic between human and machine can be very complex. Our manuscript invites a design perspective that aims to understand these dynamics and use them to improve outcomes. While we acknowledge the limitations of describing human behaviour as black-box functions, the crucial point, explained in Appendix B.1, is that from the perspective of a developer, it is natural and common to view it in this way. If a human interacts with a computer, their inputs will computationally be treated the same as the output of some function call. While human decision-making is highly complex, the developer can only use the information the human provides, which to them is the output of some function (in the mathematical sense).
>
> Continued in next comment...

---

> > ### Author Response · Authors · 2025-11-21
> > **Review response 2/4**
> >
> > ## W4 Limited empirical validation.
> > With regards to your comments about empirical validation of results and associated experimentation, we agree that further empirical work studying the effectiveness and shortcomings of various real-life HITL setups is very much needed. Ours is a theoretical manuscript, with the goal of trying to build a common language for researchers and developers to use. One of the key aspects of our work is the “Formalisation” of human-in-the-loop, which we believe is a necessary first step before any empirical work can be done. We have aimed to address complicated socio-technical questions we believe are highly important for the broader ML community who can then take this formalisation and apply it and  carry out empirical studies from it. Moreover, we have since extended or added several practice-oriented Appendices; B.9 (on practical limits of determining HITL setups and how to overcome those) has been extended, D.4 (on ML tools that can be used to aid in the creation of HITL setups) had been added, and D.6 (on the formalisation of other reductions as HITL setups) has been added. Together, we hope these now provide a better idea for how to carry out such empirical studies.
> >
> > ## W5 Abstract ADMS.
> > With regards to your comment about our use of the concept of abstract ADMSs, our considerations in this manuscript are general and not bound to particular classes of ADMS, and therefore in particular apply to machine learning based systems, as well as many others. We have chosen to use the term ‘ADMS’ in order to stress the generality of the approach and the point that we refer not to algorithms or models, but to decision-making architectures (which, naturally, include machine learned architectures). Given the rapid advancements made in the research and development of ML systems, we think that such generality is needed to develop a theoretical approach which is both applicable today and future proof.
> >
> > Continued in next comment...

---

> > > ### Author Response · Authors · 2025-11-21
> > > **Review response 3/4**
> > >
> > > # Responses to questions.
> > >
> > > All your questions raise important issues about the practical implementation of our theoretical contributions. As mentioned earlier, ours is a theoretical manuscript, aimed at framing and formalising HTIL setups to create a common language and tool for those in the ML community to build upon. As such, implementation aspects were not the primary focus, but we anticipate that our theoretical contributions can be useful in further work on implementation. With this in mind, we have now added several new Appendices (D.4 and D.6), as well as an extension to B.9, which explicitly cover implementation aspects of our analysis. In particular, Appendix B.9 now presents some approaches for how existing paradigms for documentation, testing, and auditing can be extended to HITL systems, which may be elaborated upon in future work. The overall argument in B.9 is that while auditing an ADMS with regards to the type of involvement of the human is highly complex in general, this can be greatly mitigated if the system is properly documented, and moreover developed with the facilitation of auditing in mind.
> > >
> > > ## Q1 How to test whether a system is actually an involved interaction?
> > > This is an important question to consider, and so Appendix B.9 has now been significantly extended to discuss how (developers and checkers) might check if a HITL setup is, say, an involved interaction. We do this by proposing a regulatory and cultural environment whereby developers must develop their ADMS in a way such that the extent of the human input/queries can be (formally) demonstrated/verified to any such checker. This includes rigorous documentation, and even logging methods for inputs. These ideas build on existing, established testing mechanisms for systems and software. While the Halting Problem is undecidable, there are ways around checkers being presented with an “arbitrary piece of code” to check. Adopting these mechanisms should make it possible to ascertain the degree of HITL involvement.
> > >
> > > ## Q2 How can involved interaction be made practical at scale?
> > > Scaling is indeed a valid concern for large systems, so we have now added Appendix D.4, which covers paradigms such as Rejection Learning, Learning to Defer, and Conformal Predictions. These are learning techniques which help “highlight” what, and how, and ADMS will query a human. We discuss how these can be incorporated into involved interaction setups, to help make the process more efficient and thus scale more effectively.
> > >
> > > ## Q3 Do you envision technical audits?
> > > As now discussed in the extension to Appendix B.9, we believe digital tools will greatly assist checking and auditing processes for HITL setups, just as they do for regular programmes, and we describe some of these tools there. However, as further outlined in Appendix B.9, we argue that a fully automated checking or auditing process is not only unlikely (due to issues such as the oracle problem, whereby LLMs produce outputs that are not checkable against one subjective standard), but also not desirable. The tools we describe necessarily rely on human participation, and in agreement with the rest of the manuscript we advocate that better outcomes can be achieved with human-machine interplay, rather than just one of these acting on their own.
> > >
> > > ## Q4 How does your framework prevent systems from adding meaningless queries?
> > > Restricting to “meaningful” queries is something we feel is important. And so in Appendix B.9 we now give a thorough treatment of how to start addressing and (practically) determining the types, frequency, and impact, of the queries to the human. There, we explore new requirements on developers to create, and document, their ADMS in a way that can be meaningfully checked for such a potential for “meaningless queries”. And, in Appendix D.4, we explore how to use existing tools such as Learning to Defer, that can be used by developers to help their ADMS “ask good questions” to the human. In addition, Appendix D.6 has now been added, which contains aspects about this in terms of picking the right granularity of oversight in the setup. Overall, we believe it is feasible for developers who have implemented a genuine involved interaction to convince checkers or auditors of this, while the same will be hard for developers who have implemented a system that in reality only involves barely more than endpoint action.
> > >
> > > Continued in next comment...

---

> > > > ### Author Response · Authors · 2025-11-21
> > > > **Review response 4/4**
> > > >
> > > > ## Q5 How to determine the minimum necessary level of "meaningful" human involvement?
> > > > With regards to your comment about creating HITL setups with the minimal human involvement to be legally and ethically “meaningful”, our view is that such concepts are not well defined mathematical objects. As such, and as argued in Appendix B.9, it might not be a well-posed problem to interpret what “minimal” might mean here. Rather, and as argued in B.9, “minimum standards” for legal compliance are difficult to achieve in practice, as such matters are often highly contextual. Instead, “best practice” regulatory guidance can be given to developers to help them meet regulatory requirements. We argue that developers should not aim to involve the human as little as possible, but rather consider what value a HITL can bring to their computation under which circumstances, and then design the ADMS in such a way that empowers the human to contribute this value. If the decision is in a high-risk domain or pertains to personal data, it is necessary for the ADMS to be at least as strong as endpoint action in order to satisfy the legal requirements of the EU AI Act and the GDPR. We hope that additional cases in the courts  will further refine what it means for human oversight to be ‘meaningful’ and ‘effective’ in ADMS (beyond endpoint action as established in the SCHUFA case) which we can use to develop our formalisation of HITL setups.

---

### Official Review · Reviewer_96dd · 2025-10-29

**Soundness:** 2
**Presentation:** 3
**Contribution:** 2
**Rating:** 4
**Confidence:** 2

**Summary:**

The work studies how to formalise, from a computability-theory point of view, various forms of human-in-the-loop (HITL) approaches. Then, the authors propose a taxonomy to categorize how HITL can fail, highlighting a trade-off in terms of attribution of legal responsibility and technical explainability.

**Strengths:**

The paper's main strengths are:

1. the paper presents a legal perspective for the study of Human-In-The Loop (HiTL) approaches. The perspective is relevant, as Machine Learning (ML) models are widespread to support decisions and correctly framing the responsibility for wrong predictions is very important.
2. The paper is thought-provoking, offering an interesting approach for addressing the responsibility of the actual decision. I appreciated (even if I do not completely agree) the problem of scapegoating single humans for machine failures.
3. I appreciated the focus on explainability not only on ML models, but on the entire decision-making pipeline to ensure that every decision step is transparent. I agree that this is very relevant and important to properly address the responsibilities of various agents.

**Weaknesses:**

I leave here a few open points I think the authors should discuss:


1. While I appreciate the theoretical formalization of HiTL, I would have liked the authors to offer some operational criteria to tackle the highlighted shortcomings of HiTL approaches. In the current status, it doesn't seem easy to translate the theoretical contribution into a real-life workflow.
2. I think the authors have not properly placed their work in the current literature. Several paradigms in ML aim to involve humans in the loop, such as learning to defer (see e.g., [1]), machine learning models that abstain (see e.g., [2]) and decision support systems through conformal prediction (see e.g., [3]). Discussing these more recent lines of work is, in my view, mandatory.
3. I think the authors should explicitly state why they are focusing only on the EU/UK environment. Enlarging to other countries/law frameworks could be impactful and provide a valuable picture of HiTL's existing legal frameworks worldwide.

[1] Madras, D., Pitassi, T., & Zemel, R. (2018). Predict responsibly: improving fairness and accuracy by learning to defer. Advances in neural information processing systems, 31.

[2] Ruggieri, S., & Pugnana, A. (2025). Things machine learning models know that they don’t know. In Proceedings of the AAAI Conference on Artificial Intelligence (Vol. 39, No. 27, pp. 28684-28693).

[3] Straitouri, E., Wang, L., Okati, N., & Rodriguez, M. G. (2023, July). Improving expert predictions with conformal prediction. In International Conference on Machine Learning (pp. 32633-32653). PMLR.

**Questions:**

Please discuss the highlighted weaknesses.
Regarding W1, I think adding some examples of countermeasures that could have been implemented in the Uber case would help.

---

> ### Author Response · Authors · 2025-11-21
> **Review response 1/2**
>
> We thank the reviewer for their valuable feedback on operationalising our formalism and better connecting it to the wider ML literature, and for asking about a stronger justification of the jurisdictional focus of this manuscript . We have incorporated all three concerns by extending our discussion in appendix B.9 (on determining, testing and documenting HITL types) and by adding two new appendices (D.1 on our choice of legal jurisdictions to cover, and D.4 on HITL and its connection to different learning strategies)
>
> # Responses to weaknesses.
>
> ## W1 Turning theory in practice.
> We agree that verifying a HITL setup should not be viewed as a purely technical problem, but rather as a sociotechnical, practice-oriented task. To address this, we have substantially extended Appendix B.9 and its discussion of the technical, legal, and moral challenges to now include some concrete ideas for how our theoretical framework may be used in practice.
>
> Specifically, we argue that the burden of proof for verifying a HITL type should shift partially onto developers, and not just lay with auditors or other external checkers. We now discuss some suggestions to achieve this, including:
> - Enhanced documentation: We discuss extending existing frameworks, such as model cards and datasheets, to explicitly document the HITL reduction type and the nature of human queries.
> - Holistic oversight: We make the connection to Manheim & Homewood’s (2025) five-stage AI supervision maturity model to assess how a team and organisation engages with oversight risks.
> - Formal proofs: We briefly discuss the potential to have formal verification/proofs of specific HITL types.
> - Crowd-based Testing: We connect our work to the HINT (Human-AI INtegration Testing) methodology to empirically sample data about how humans perform within a specific HITL setup
> - Verification: We propose reverse checks (counterfactual explanations) and specific logging procedures to verify that human inputs are computationally meaningful.
>
> In addition to this, we have now supplemented our Failure Taxonomy (Section 3.1) and worked example of a failed HITL setup (Uber case of a self-driving car, in Section 3.3) by adding Appendix C.5 which applies our HITL failure taxonomy to show how Uber (and others) could have done things differently to avoid the fatality from the self-driving car HITL setup. Furthermore, our existing discussion of the SCHUFA case and how SCHUFA could have used an involved interaction to prevent the endpoint action HITL setup slipping back to trivial monitoring (covered in Section 4.2 and associated Appendix D.3) gives another such example of our our theoretical contribution being directly used to change and improve an existing workflow. We very much agree that laying out theory is only part of the process, and as such have shown how to make active use of our theory to “do things better” and produce more effective, and ultimately safer, HITL setups.
>
> ## W2 Missing literature and connection to learning strategies.
> We agree that learning to defer (L2D) and other learning strategies are important aspects of building a HITL setup in practice. However, we also note that many of the mentioned strategies (L2D, and Rejection Learning) do not automatically lead to a HITL setup as we have formalised it in this manuscript, nor to “meaningful human oversight” as per the UK/EU GDPR or EU AI Act. To address this, we have added a new Appendix (D.4) which explicitly analyses the connection between our formalisation and the suggested literature.
>
> Specifically, while the standard presentation of L2D and rejection learning allows a machine to exclude a human from the loop (thus turning the setup into a HOOTL), we see L2D as a powerful tool for operationalising an involved interaction. Taking a holistic perspective of L2D, we decided to discuss both benefits and drawbacks. Crucially, we now discuss how L2D can help to deal with complexity, handle high-volume/high-speed queries, manage human load (cognitive, and work), and create a setup wherein the human is asked a query when it can be (meaningfully) expected that they can contribute something important. In addition, the discussion in Appendix D.4 now also includes the danger of eliminating human agency, the potential for legal non-compliance, and the introduction of alignment issues.
>
> In Appendix D.4 we also cover conformal predictions, looking at how as a standalone HITL setup they constitute an endpoint action under our formalisation. We also discuss how this can be used within a much larger involved interaction HITL setup, reducing the cognitive load on the human by presenting them “slimmed-down” sets of responses to choose from for a given query. While this then raises automation bias and human agency issues in itself, it nonetheless presents a trade-off that developers can consider when designing their involved interaction setup.
>
> Continued in next comment...

---

> > ### Author Response · Authors · 2025-11-21
> > **Review response 2/2**
> >
> > ## W3 Focus on EU/UK law.
> > We thank the reviewer for asking for further justification for our jurisdictional focus. We agree that this should be fully explained in the manuscript. Our manuscript shows, as proof of concept, how mathematical formalism, failure modes, and law can come together to create a coherent system to create and analyse HITL setups. The manuscript focuses on EU and UK laws for three reasons:
> >
> > 1) The “Brussels Effect”: the EU AI Act is the world’s first comprehensive approach to regulating AI. We argue that, similar to the EU GDPR before, it can be reasonably expected that it will serve as an inspiration for other jurisdictions.
> >
> > 2)  The UK’s common law of negligence acts as persuasive authority in Commonwealth jurisdictions, such as Australia, Canada, India, and South Africa. Our analysis of case law on causation and HITL failures is therefore not only limited to the UK, but is applicable more widely across the globe.
> >
> > 3) While the US/China are forerunners in AI development and therefore crucial from a legal perspective, they present additional challenges which warrant a separate dedicated legal analysis: the US landscape comprises a patchwork of rules through a complex web of state and federal law, and China often uses a targeted approach for specific products and services.
> >
> > We added a full justification of our scope of analysis in Appendix D.1.

---

> > > ### Comment · Reviewer_96dd · 2025-11-23
> > >
> > > I thank the authors for their answer, which has clarified my concerns. Hence, I am increasing the score.

---

> > > > ### Author Response · Authors · 2025-11-26
> > > >
> > > > We thank the reviewer for engaging with us, and are happy to answer any further questions they might have.

---

### Official Review · Reviewer_gJTx · 2025-11-02

**Soundness:** 3
**Presentation:** 4
**Contribution:** 4
**Rating:** 10
**Confidence:** 3

**Summary:**

This paper is an attempt at formalising several aspects of the human-in-the-loop (HITL) paradigm for automated decision making systems (ADMS).

The contributions of the paper can be summarized into three main points:
- A formalisation of different HITL modes in terms of oracle machines.
- A taxonomy of failure modes in HITL systems.
- A discussion about responsibility and accountability of HITL systems.

The paper defines human-in-the-loop systems as oracle machines, where the human is the oracle, and proposes three main categories depending on the level of involvement of the human: monitoring, endpoint, and involved interaction. In monitoring (called in the paper *human trivially monitoring the loop* or *trivial monitoring*), the only intervention that the human can have is to halt the execution of the ADMS. This models, for example, systems in which a human observer can stop the ADMS and take control as an emergency response. In endpoint (in the paper *human at the end of the loop* or *endpoint action*), the human receives a single query, and the program halts after the human's response. This models systems in which a program presents a set of solutions or courses of actions to a human, and the human makes the final decision. In involved interaction (also called in the paper *human involved in the loop*), the machine can ask an unbounded amount of queries to the human, and these must have a signficant effect in the resulting computation.

The paper classifies HITL failures into five categories (failure of the machine, failure of the human, failure of the human-machine interface, failure of the human-machine workflow, and failure due to exogenous circumstances), and discusses the connection of the failure modes with the HITL categories. The different failure modes are also illustrated through a real-world failure example, in which a HITL autonomous car was involved in a fatal traffic accident.

Finally, the paper discusses the legal frameworks, mainly in the EU and UK, as they pertain to HITL systems, and the tradeoffs in terms of assigning responsibility between the human and the machine.

**Strengths:**

S1. Relevant and timely topic.

S2. The paper is clearly structured and very well written.

S3. The formalisation in terms of oracle machines is well inspired. The illustration of different failure modes in the different examples (one in the main text, two in the supplementary material) is convincing.

S4. The paper engages with relevant legal frameworks (in the EU and the UK), analyses the aspects relevant to HITL and proposes improvement points to avoid real problems that have arisen from the use of HITL in safety-critical applications, like the use of the human in the loop as a scapegoat to avoid responsibility by the ADMS deployer.

**Weaknesses:**

I don't see major weaknesses. As minor points:

**W1.** The paper leaves a lot of the content in the appendix, and the many pointers to the appendix in the main text are distracting. As a reader, it is not clear when I will be able to follow the paper just from the main text, and when I need to go and read the supplementary content.

**W2.** The choice of limiting HITL paradigms to three concrete ones, leaving many intermediate steps unformalised, is a weak point of the paper. I know this is a conscious choice motivated in the text. However, it seems that the discussion, particularly in what regards the EU and UK legal frameworks, would need to include these paradigms with intermediate involvement of the human.

**W3.** The halting abilities of the human are not completely clear.

- In trivial monitoring, it is not clear whether the human has the ability to halt the ADMS at any time, or only when it asks a query. The former would not be realistic in certain situations, and the latter would then require the nuance of when does the machine query the human as part of the analysis.
- In endpoint interaction, it is not clear whether the human has the ability to halt the machine before it reaches the endpoint query.

**W4.** In lines 349 to 362, you argue that trivial monitoring cannot be seen as the *meaningful oversight* required by the EU GDPR and AI Act. I think the deliberate choice of words (like *trivial*) and the lack of specificity in the halting capabilities of the human (see W3) are playing a hidden but important role in this argument. To take an example from this paper, in the Uber case, the car has a trivial monitoring HITL setup, in which the human can resume control of the vehicle at any time with very low friction (just touching the steering wheel or the pedals). I doubt that the EU law requires more involvement of the human to be considered *meaningful oversight*. While I'm not aware of levels of autonomy as in the Uber case being currently deployed in the EU, classical cruise control systems in cars are very common, definitely within legal margins, and offer no more than trivial monitoring by the human in the loop. Consider maybe not using the word *trivial* in your taxonomy.

**Questions:**

**Q1.** The difference of failure mode 2 (process and workflow) and failure mode 4 (failure of the human) is not clear to me. Can you provide examples where they are independent failure modes? That is, and example in which there is a failure of the process without a failure of the human, and an example in which there is a failure of the human without a failure of the process.

Q2. In lines 1151 to 1158, you argue that deciding whether an HITL system is actually of the involved interaction type from a moral perspective is particularly difficult because of two reasons: the machine may ignore the human's input, and the machine may act contrary to the human's input.
- **Q2.1** For the first case, how is this different from the machine not issuing a real query? Shouldn't this case already be covered by the definition of involved interaction as requiring real queries?
- **Q2.2** A machine could be designed to act against the human inputs and still be moral. For example, a human-like robot used to train humans in conflict resolution. Would this be accounted by your framework?

---

> ### Author Response · Authors · 2025-11-21
> **Review response 1/3**
>
> We thank the reviewer for their nuanced and meaningful reading of our submission, and subsequent suggestions. We were happy to carry out further investigation of how to formalise and analyse intermediate setups as HITL reductions (done in Appendix D.6), and to highlight the distinctions between failure modes 2 and 4 in our taxonomy (done in Appendix C.5); both of these are useful additions to the manuscript. The suggestions to clarify the use of the term “trivial”, what the halting abilities of the human are, and how appendices are used, have helped improve the readability of the manuscript. And the feedback about considering dealing with a machine going against human input helped us to expand on the first part of Appendix B.9 to address this important concern.
>
> # Responses to weaknesses.
>
> ## W1 Pointers to the appendix.
> We agree that signposting could be improved. Hence, we have made the effort to improve the signposting to the appendices, and now differentiate between “supporting details” and “further discourse” depending on how essential an appendix is to the overall argument.
>
> ## W2 Intermediate reductions in legal contexts.
> We agree that there are many reduction options between our primary considerations. Therefore, we added a new Appendix (D.6) on making use of intermediate reductions in HITL setups, and how our existing mathematical, practical, legal, and moral analysis applies to those setups. This shows how, legally, the laws we consider do not differentiate between any of the setup types beyond endpoint actions (including intermediate reductions) for oversight requirements, but for explainability requirements, the setups form a “sliding scale”, allowing developers to prioritise explainability over neat attributions of responsibility depending on the context. We also show how one might mathematically formalise HITL setups from this (or indeed any) set of reductions.
>
> ## W3 Halting abilities of the human.
> We agree that being clear about the human's halting abilities is important. Therefore we've made the following changes: In Section 2.1, we now clarify the halting abilities of the human in each of the HITL setups described. To clarify: in all such HITL setups, the human can halt the machine at any point in the computation, whether invited to or not. We note that, of course, practical limitations exist which means this is not a perfect process; such limitations are covered in Section 3.1.
>
> ## W4 Can “trivial” monitoring be meaningful oversight?
> We thank the reviewer for raising concerns about using the word “trivial”, and for asking for further clarification. Responding to this, in Section 2.1 we now clarify that our use of “trivial” refers only to the computational properties. We have decided to retain the terminology of “trivial”, for the following reasons: these setups are “trivial” because there are no real queries exchanged between the oracle machine and human. The human’s monitoring position is “trivial” because no computational intervention is intended. Nevertheless, a trivial monitoring setup can still provide effective practical interventions in certain situations. Since the scope of the EU AI Act and GDPR human oversight requirements only operate in high-risk scenarios, the use of the term “trivial” might not seem appropriate. However, we believe our notion of trivial monitoring corresponds to non-effective human oversight as per Article 14 of the EU AI Act and would not “break automation” as per Article 22 of the GDPR which prohibits fully automated ADMS. Therefore we deliberately use “trivial” in these high-risk settings as a way to communicate that the system described does not fulfill the necessary purpose or safety requirements in accordance with the law.
>
> We also thank you for your question regarding self-driving cars and the EU AI Act. Having looked into your comment about EU law and meaningful oversight of a self-driving car (as in the Uber case), our understanding of such law (the EU AI Act) is that it would indeed view the role of the human in that car as not having “effective oversight”. Please see our final comment at the end of this chain for more details.
>
> Continued in next comment...

---

> > ### Author Response · Authors · 2025-11-21
> > **Review response 2/3**
> >
> > # Responses to questions.
> >
> > ## Q1 Difference between failure mode 2 and failure mode 4.
> > We appreciate that failure modes 2 and 4 in our taxonomy might seem similar. To clarify the distinction between these, we added a new Appendix (C.2) which explicitly demonstrates the distinction between failure modes 2 and 4. The appendix explains the difference by considering a situation where the processes failed, but the human component worked (the 1983 nuclear false alarm incident) and where the human component failed, but the process was not a problem (a GPS incident, where a human failed to scrutinise an algorithmic instruction and drove into the ocean).
> >
> > ## Q2 Machines going against the input.
> > This is an interesting question on many levels (philosophical, regulatory, ethical, and technical). These concerns are now addressed in the first half of what is now Appendix B.9 (which is where the issues originally arose). In the case where the machine “ignores the input”, this would indeed not be a real query, but potentially masked as one seeing as the machine did ask, and could have acted on, that query. In Appendix B.9 we further discuss how, if a machine were to intentionally go against a human response, while this would constitute a real query, without additional human oversight (potentially from a different human) it would indeed violate the concept of human agency.
> >
> > Continued in next comment...

---

> > > ### Author Response · Authors · 2025-11-21
> > > **Review response 3/3**
> > >
> > > # Comment regarding W4 and trivial monitoring
> > >
> > > As a follow-up about Uber and meaningful control, we first wish to clarify that the decision in the Uber case was based on US criminal laws (not the EU AI Act or GDPR), and is only used in this manuscript as an example of HITL setup failure and subsequent scapegoating. However, we can apply the EU AI Act requirements on human oversight and the HITL failure modes outlined in this manuscript to this example. The setup here fits into the class of trivial monitoring which we argue is not enough to comply with the EU’s requirements for “effective oversight” as per Article 14. Article 14(4)(a) requires that the human is able to “duly monitor its operation, including in view of detecting and addressing anomalies, dysfunctions, and unexpected performance”. Uber required the human user to perpetually “hover their hands above the steering wheel and foot above the brake pedal” to monitor the system. Aside from the unrealistic expectations of this, the human driver had no access to information, like uncertainty scores when the machine was classifying objects. Thus the involvement of the human was too weak to actually make the monitoring “effective” and capable of helping the driver detect or address “anomalies, dysfunctions, and unexpected performance”. In addition, Article 14(4)(b) requires deployers to remain aware of automation bias in light of the vast evidence that humans over-rely on ADMS’ outcomes. This includes the incentive structures around the human overseer, which Green (https://www.sciencedirect.com/science/article/pii/S0267364922000292 , pg 6)  argues requires that the human can “thoroughly consider all of the information relevant to a given decision” which includes, for instance, having enough time to make informed choices (https://doi.org/10.1002/poi3.198 , pg 19). As stated in Section 3.3, the human driver received a delayed notification warning of the upcoming impact.
> > >
> > > A poor safety culture was also identified and reported during the rollout of the technology. There is uncertainty about how the human oversight provisions in Article 14 of the EU AI Act will be implemented in practice, but in accordance with our strict reading, the trivial monitoring in the Uber case would not be considered “effective” oversight. In the appendix C.5, we include some ways in which the failure modes could have been used to make the oversight more effective as per the requirements in Article 14.  This could  include some change point detection within the classification system to trigger a warning to the Uber driver to increase vigilance, enforce safety training work and perform regular checks of work; the setting of human periodic tasks to reduce human fatigue (for more in depth discussion, see C.5).
> > >
> > > It is true that classical cruise control systems are not governed by the same requirements. One reason for this might be because when a car is in cruise control, the human still retains control over the car (e.g., by steering) – the same is not true with a self-driving car where the human driver has no control until they take the wheel. It has been reported that the time for drivers to gain full control of self-driving cars is also time consuming, taking around 10 seconds to transition from an “automated state” (https://doi.org/10.1002/poi3.198 , pg 109). However, another important distinction is that self-driving cars use AI, in the sense that they are automatically governed by the EU AI Act (definition of AI in Article 3(1) “a machine-based system that is designed to operate with varying levels of autonomy and that may exhibit adaptiveness after deployment, and that, for explicit or implicit objectives, infers, from the input it receives, how to generate outputs such as predictions, content, recommendations, or decisions that can influence physical or virtual environments”). Self-driving cars also fall within the category of high-risk AI systems in Article 6(2) as per Annex III (critical infrastructure, road traffic), subjecting them to the highest requirements in the Act. EU law has taken the approach to regulate AI differently than other technologies. It is beyond the scope of this manuscript to consider the normative arguments relating to this question, though further research could focus on this.

---

> > > > ### Comment · Reviewer_gJTx · 2025-11-26
> > > >
> > > > I thank the authors for their thorough response, in particular the detailed discussion of the term "trivial monitoring" and its relation to the Uber fatality case, and a potential implementation of self-driving cars under EU law.
> > > >
> > > > I see your point about "effective oversight" having strong requirements. On the other hand, a human driver has effective oversight (actually, effective control) of a car with the standard information (such as speedometer, revolutions, gas level, etc), so it sounds reasonable to me that a human that is paying attention to the road would also be considered to have "effective oversight" under article 14, even if it has no access to internal detection scores.
> > > >
> > > > To me, the Uber case underlines the issue of unrealistic expectations and human scapegoating, and begs the question of whether the human-in-the-loop model is even useful or desirable for automated driving: if we require the human overseer as much cognitive involvement as it would be required to just drive the car, what is the point of the self-driving algorithm?
> > > >
> > > > In any case, I appreciate the discussion and all the pointers to the relevant legislation.
> > > >
> > > > The only real issue I have is the reference for taking around 10 seconds to transition from an "automated state". You cite [a, page 109], which indeed has the 10 second figure, citing [b,c,d].
> > > > - As far as I can tell, [b] has no discussion of concrete reaction time, other than it is a challenge for automated driving.
> > > > - [c] reports on an experimental study where subjects are shown videos of a car simulation (from no context). While the Uber driving example has unreasonable expectations (like constantly hovering the hands over the steering wheel), it is not unreasonable to ask humans monitoring a self driving car to be paying attention to the road.
> > > > - The closest of the studies is [d], where subjects are indeed presented with a driving simulation, and an automated driving system is engaged and disengaged at different times. This study does observe a typical 10 second lag between the automated driving system disengaging and the subjects focusing on the road. However, the subjects in this study were not given any signal or notification that the automated driving system was disengaged [at least, I could not find it reported in the paper]. So I find it more likely that the 10 second lag is mostly the time it took the subjects to notice the automated driving system was off, and this reaction time would realistically be much shorter when the driver is properly notified.
> > > >
> > > > I have to say I was not familiar with this literature, so I may have missed something, but as far as I can tell, it does not support the claim of a 10 second lag to take control.
> > > >
> > > > ------
> > > > References
> > > >
> > > > [a] B. Wagner. Liable, but Not in Control? Ensuring Meaningful Human Agency in Automated Decision-Making Systems. Policy and Internet. 2019.
> > > >
> > > > [b] M. Kyriakidis et al. "A Human Factors Perspective on Automated Driving."  Theoretical Issues in Ergonomics Science. 2017.
> > > >
> > > > [c] Z. Lu et al. "How Much Time Do Drivers Need to Obtain Situation Awareness? A Laboratory-Based Study of Automated Driving". Applied Ergonomics. 2017.
> > > >
> > > > [d] N. Merat et al. "Transition to Manual: Driver Behaviour When Resuming Control From a Highly Automated Vehicle." Transportation Research Part F: Trafﬁc Psychology and Behaviour. 2014.

---

> > > > > ### Author Response · Authors · 2025-12-02
> > > > > **Further comment regarding trivial monitoring and self-driving cars**
> > > > >
> > > > > We thank the reviewer for their continued engagement with us, and for all their careful work in investigating the additional details surrounding human-in-the-loop in the context of self-driving cars. We will respond here in a somewhat reverse order, as our response on the “10 second lag” will be relevant for our discussion on Article 14.
> > > > >
> > > > > Firstly, just to clarify, in the Uber case raised in the manuscript, the human driver was put there as a safety mechanism, as the self-driving system was still in some sort of testing phase by Uber. These were tests, not real journeys. In addition, we completely agree with the reviewer that unrealistic expectations and human scapegoating raise a valid question of whether the human-in-the-loop model is a useful safety mechanism for automated driving (be it in the testing or deployment phase).
> > > > >
> > > > > With regards to our comment on the “10 seconds to transition”, we thank the reviewer for looking into this in further detail. We have now read ([d] N. Merat et al.), which we shall continue to refer to as [d] here. Our observations are as follows:
> > > > > - Just like the reviewer, we could not find any mention of an explicit “warning” (such as an alarm) to the human driver to indicate that control had been passed back to them. All we could find was the statement that “A small LCD panel below the speedometer was backlit and displayed ‘‘ACC/LKS’’ when the highly-automated system (lateral plus longitudinal control) was active.” [d, p.276]. We presume that this light would then turn off when the automated system was no longer active. Whether it was the human (eventually) noticing this light turning off, or the car starting to veer and/or slow down, that triggered the human to take the wheel, is unclear to us. We agree that the lack of obvious announcement may have greatly contributed to this 10 second delay in transition to the human driver, and a more obvious alarm may have shortened this delay.
> > > > > - However, we also note that [d] mentions “...it took drivers around 35–40s to stabilise their lateral control of the vehicle” [d, p.274], and goes on to say “Following a dramatic rise in the number of corrections after 10s, they are seen to stabilize after around 35–40s.” [d, p.280]. Indeed, one sees in figure 5 of [d] that the human driver has a period between 10s and 40s after the automated system was disengaged where they (they human) were making many more steering corrections. This period of approximately 30 seconds of what we might describe as ‘wobbly driving’ happened *after* the human had retaken control.
> > > > > - While an active warning alarm may have helped the human driver retake control faster than the 10s observed in [d], we cannot see what could have been done to avert the following 30 seconds of ‘wobbly driving’, and thus it seems that [d] provides some evidence that human drivers have an inherent transition window to *properly* retake control of a vehicle when control is handed back to them.
> > > > >
> > > > > Again, we thank the reviewer for highlighting this work to us.
> > > > >
> > > > > Thus, with regards to the comment about Article 14 and a human driver having effective oversight, it appears to us as though [d] demonstrates that there is a very real distinction between *driving a car* and *being ready to take over the driving of a car*; the 30 seconds of ‘wobbly driving’ given in [d] seem to be somewhat unavoidable. In light of this, and alongside the further discussion of all the failure modes described in Section 3 our manuscript, we maintain that a human driver passively sitting in the driver seat of a self-driving car is unlikely to have "effective oversight" under Article 14. We should note that, as far as we are aware, there has not been additional guidance on the scope of obligations imposed by this Article other than scholarly commentary that we have already cited in our manuscript and comments.
> > > > >
> > > > > To conclude, we wish to clarify that our discussion in Review response 3/3 ("Comment regarding W4 and trivial monitoring") for this reviewer, and in this response, is not currently planned to be included in the manuscript.

---

### Author Response · Authors · 2025-12-03
**Summary of the paper and reviewer-author comments**

We summarise here the reviewer and author comments, and how we have now improved some of our arguments in the paper as a result of this feedback.

\
**In summary,** our paper formalises Human-in-the-loop (HITL) setups and distinguishes 3 types: trivial human monitoring, single endpoint human action, and highly involved human-machine interaction. It also provides a taxonomy of HITL failure modes, expanding the common notion of “human error” into a myriad of factors. Combining these, we provide a legal and moral analysis of how certain HITL setups might be insufficient to address compliance and safety concerns.

\
**Reviewer gJTx** noted that the paper was timely, very well written, and gave both an inspired formalisation in terms of oracle machines, and convincing examples of failure modes. They also highlighted the paper’s engagement with UK/EU law, and its proposed improvements on HITL use in safety-critical applications such as the avoidance of human scapegoating by developers.

They sought clarity on which appendices were necessary background, which we now flag throughout the paper. They asked about the specific halting abilities of the human, which we now make explicit (Section 2.1). And they asked about the intermediate reduction types, which we now fully formalise and apply our legal and moral analysis to (Appendix D.6).

They requested examples differentiating between failure categories 2 and 4, which we now provide (Appendix C.2). They asked about the subtleties of a machine ignoring, or going against, the human's input; we now address these in Appendix B.9 with a deep discussion on human agency. And they asked whether trivial monitoring might be considered as effective oversight under EU law (in the context of self-driving cars); we defend here that it would be seen as ineffective in law, as the human has zero control before taking over, after which difficulties such as time to react and establish *stable* control arise.

\
**Reviewer 96dd** noted that the paper presented a relevant legal perspective on HITL, gave a thought-provoking approach for addressing responsibility, and placed a relevant and important focus on the entire ML decision-making pipeline.

They asked for operational criteria to supplement our theoretical formalisation, which we now provide (Appendix B.9), detailing actions developers can take such as enhanced documentation, holistic oversight, formal proofs, crowd-based testing, counterfactual verification, and logging. They also asked how our framework could be implemented in the case study on self-driving cars, which we now make explicit (Appendix C.5), detailing how the failures in that example could have been averted through use of our taxonomy.

They asked for connections to existing ML paradigms such as learning to defer, rejection learning, and conformal predictions; we now give details on how these can be used within our formalised HITL setups (Appendix D.4). And they (and reviewer LN6b) asked about the focus on EU/UK regulatory environments, which we now fully justify (Appendix D.1) with connections to the Brussels Effect and the global impact of UK Common Law.

\
**Reviewer LN6b** noted the paper gave interesting case studies bridging theory to incidents, a novel formalisation of HITL systems that is creative and mathematically sound, a concrete reading of the GDPR/EU AI Act, a trade-off analysis of explainability vs. responsibility with policy remedies, and overall was an engaging read and worth thinking more about.

They asked how auditors might check for things such as “real human queries”, which we now cover in Appendix B.9 with regulatory obligations on developers to build ADMSs where the extent of human input can be (formally) demonstrated. They asked about technical audits of HITL types; we now detail (Appendix B.9) digital tools which assist human auditing. They also asked how to efficiently scale involved interactions, which we now describe in Appendix D.4.

They asked how to prevent systems adding meaningless queries, which we now cover with methods for determining query types (Appendix B.9), development tools to ensure “good” querying (Appendix D.4), and ways to choose the right granularity of human oversight (Appendix D.6). They asked how to determine minimum levels of human involvement to remain “meaningful”; we now argue in Appendix B.9 why “best practice” regulatory guidance is preferable to “minimum standards”.

They asked about the limits of viewing humans as oracles; we re-iterated to them that an ADMS can only use the information the human provides, which, computationally, acts as some (oracle) function, as already covered in Appendix B.1. And they queried the levels of empirical validation provided; we outlined to them that further empirical work is needed, but that ours is a theoretical paper aimed at building a common language through formalisation before empirical studies can be carried out.

\
*This summary was written without the aid of AI.*

---

### Meta-Review · Area_Chair_Kqj3 · 2026-01-08

**Summary:**

Reviewers raised concerns pertaining to: ambiguity in HITL definitions and halting abilities; intermediate HITL configurations and legal applicability; failure mode taxonomy; conceptual clarity regarding whether systems that override humans undermine the notion of ``involved interaction’’; operationalization and insufficient guidance for developers, auditors and regulators; missing literature; as well as jurisdictional focus.

This is a novel, rigorous and timely contribution and the authors satisfactorily and thoroughly address almost all concerns raised. The authors provide extensive and high-quality revisions adding multiple appendices and clarifying scope, intent and limitation. See below for further details

**Reviewer Concerns:**

The authors formal clarification added across the main text and appendices addresses the ambiguity in HITL definitions and halting abilities concern. The intermediate HITL configurations and legal applicability concern has also been fully addressed with the new added appendix D.6 that formalises intermediate reductions plus, they also demonstrate that their framework generalizes cleanly while preserving legal analysis.

The added appendix with concrete, historically grounded counterexamples satisfactorily clarified independence of failure categories.

Lack of conceptual clarity regarding whether systems that override humans undermine the notion of ``involved interaction’’ was addressed by the extended appendix and clear distinction made between fake queries, masked queries, and morally permissible counter-human behavior with added oversight.

Concerns regarding operationalization and insufficient guidance for developers, auditors and regulators were addressed with extensions added to appendix B.9 and newly added appendix, now D.4. These additions clarify documentation requirements (model cards, HITL declarations), logging and reverse checks and shared burden between developers and auditors as well as providing concrete counterfactuals showing how designs could have been changed

missing literature pertaining to learning to defer, abstention, etc was also fully addressed and the added appendix clarifies why they focus on EU and UK only.

**Reviewer Scores:**

This paper received mixed ratings: 1 x 10 (strong accept), 1 x 4 raised from a previously lower Rating (marginally below the acceptance threshold. But would not mind if paper is accepted), and 1 x 6 (marginally above the acceptance threshold)

---

### Decision · Program_Chairs · 2026-01-26

Accept (Poster)